

# An updated emission inventory of vehicular VOCs/IVOCs in China

Huan Liu[1,2,*], Hanyang Man[1,2], Hongyang Cui[3], Yanjun Wang[4], Fanyuan Deng[1], Yue Wang[1], Xiaofan Yang[1], Qian Xiao[1], Qiang Zhang[3], Yan Ding[4], Kebin He[1,2]

[1]State Key Joint Laboratory of Environment Simulation and Pollution Control, School of Environment,

Tsinghua University, Beijing, 100084, China

[2]State Environmental Protection Key Laboratory of Sources and Control of Air Pollution Complex,

Beijing, 100084, China

[3]Ministry of Education Key Laboratory for Earth System Modeling, Center for Earth System Science,

Tsinghua University, Beijing, 10008, China

[4]Vehicle Emission Control Center (VECC) of Ministry of Environmental Protection, Beijing, 100084,

China

*Correspondence to*: Huan Liu (liu_env@tsinghua.edu.cn)

**Abstract.** Currently, the emission inventory of vehicular volatile organic compounds (VOCs) is one of those with the largest errors and uncertainties due to the imperfection of estimation methods and the lack
of first-hand basic data. In this study, an updated speciated emission inventory of VOCs and an estimation of intermediate-volatility organic compounds (IVOCs) from vehicles in China at the provincial level, with a target year of 2015, were developed based on a set of state-of-the-art methods and a mass of local measurement data. The activity data for light-duty vehicles were derived from trajectories of more than 70 thousand cars for one year. The annual mileages of trucks were calculated from reported data by more
than 2 million trucks in China. The emission profiles were updated using measurement data. Not only vehicular tailpipe emissions (VTEs) but also four kinds of vehicular evaporation emissions (VEEs), including refuelling, hot soak, diurnal and running loss, were taken into account. The results showed that the total vehicular VOCs emissions in China were 4.21 Tg and the IVOCs emissions were 121.23 Gg in the year of 2015. VTEs were still the predominant contributor, but VEEs were already responsible for
39.20% of VOCs. Since VEES has a much less strict control standard, it should be paid much more attention to. Among VTEs, passenger vehicles contributed most (49.86%), followed by trucks (28.15%) and motorcycles (21.99%). Among VEEs, running loss was the largest contributor (81.05%). For both VTEs and VEEs, Guangdong, Shandong and Jiangsu province took the first three spots, with a respective contribution of 10.66%, 8.85% and 6.54% to the total amounts of VOCs from vehicles. Totally, 97 VOC
species were analysed in this VOCs emission inventory. I-pentane, toluene and formaldehyde were found to be the most abundant species in China's vehicular VOCs emissions. The estimated IVOCs is other



'inconvenient truth', providing insights the precursors of secondary organic aerosol (SOA) from vehicles were much more than previous estimation.

## 1 Introduction

China is one of those countries that are facing severe threats of $PM_{2.5}$ (particulate matter with aerodynamic diameters of less than 2.5μm) and ozone pollution at the same time (*Sitch et al., 2007; van Donkelaar et al., 2015*). In the year of 2010, nearly 1.36 million premature deaths in China were attributed to these two major types of chief pollutants (*Lelieveld et al., 2015; Liu et al., 2015*). Studies on $PM_{2.5}$ pollution indicated that secondary organic aerosols (SOA) accounted for a significant proportion of ambient $PM_{2.5}$ mass in Chinese cities (*Cui et al., 2015*). Recent studies suggested that both volatile organic compounds (VOCs) and intermediate-volatility organic compounds (IVOCs) contributed to SOA formation. IVOCs is a series compounds with effective saturate concentration between $10^3$-$10^6$μg/m$^3$, corresponding to the volatility range of C12-C22 n-alkanes (*Zhao et al., 2014*). In some regions, IVOC could be dominant (*Huang et al., 2014; Robinson et al., 2007; Hodzic et al., 2010*). Studies on ozone pollution also demonstrated that ozone formation was controlled by VOCs in many major Chinese cities (*Tie, et al., 2006; Geng et al., 2008; Shao et al., 2009*). Undoubtedly, to achieve better air quality and to reduce the health hazards resulted from air pollution in China, VOCs and IVOCs should be paid great attention to.

Previous studies have repeatedly reported that, among the various anthropogenic emission sources in Chinese cities, vehicles were the predominant contributor to both VOCs emissions and ambient VOCs concentrations (*Song et al., 2008; Zheng et al., 2009; Wang et al., 2010; Shao et al., 2011; Cui et al., 2015*). A comprehensive and accurate national emission inventory is critical to the design of effective abatement strategies on pollution control at the country level. Cai et al reported VOCs emission inventory from on-road vehicles in China 1980-2005 (*Cai et al., 2009*). Several other studies also include vehicles as part of the transportation section in their comprehensive emission inventories of VOCs (*Tonooka et al., 2001; Klimont et al., 2002; Streets et al., 2003; Bo et al., 2008; Liu et al., 2008; Wei et al., 2008; Zhang et al., 2009; Cao et al., 2011; Zheng et al., 2014*). The complete summary of existing studies on vehicular VOCs emission inventory and their respective performance were shown in Table S1 in the Supporting Information. These existing emission inventories have greatly improved our understanding





on VOCs emissions. We noticed that all the studies above provide emission inventory before 2010 and none of the previous studies took IVOCs into consideration. Considering the dramatical increase trend of vehicle population, it is extremely urgent to establish new emission inventories. However, there were multiple key factors changed in the last ten years which require new method and data.

First of all, dominant VOCs emission processes of vehicles may switch. Compared with vehicular tailpipe

emissions (VTEs), vehicular evaporation emissions (VEEs) has been proved to be a non-ignorable contributor to the ambient VOCs concentrations recently (***Yamada et al., 2013; Liu et al., 2015***). VOCs evaporate from gasoline fuelled vehicles continually whether they were refuelling, running or parked. The vapors generated go through the equipped carbon canister and eventually into the ambient atmosphere. Besides, they could also permeate through elastomers of the vehicle's fuel system to enter the atmosphere

at the same time. According to the state vehicles are in, evaporative emissions come in four varieties: refuelling loss, running loss, hot soak loss and diurnal loss. To include VEEs in the inventory, local emission factors and profile of VEEs are necessary because they are highly related with local gasoline formula and vehicle controls. In addition, a more sophisticated method is necessary to estimate VEEs. The details were further described in method section.

Secondly, as a great contributor to the formation of SOA, IVOCs have strong impact on atmospheric condition, global climate and human health. However, there are few studies on IVOCs because of the complicate composition of IVOCs. For these category components with long chains, short of systematic and integrated analysis methods limits the progress of measuring and quantifying IVOCs (***Goldstein et al., 2007; Jathar et al., 2014***). Therefore, only few studies provided emission measurements results of

IVOCs, none of the IVOCs emission inventory has been reported.

Most importantly, big data on vehicle activity may greatly reduce the uncertainty of emission inventory. Vehicle activity is critical to total emission estimation. In previous studies, these parameters were usually from hypothesis based on experience from other countries, or surveys from limited samples (usually less than 2000) (***Liu et al., 2007; Yang et al., 2015***). With the development of transportation networking

technology, we were able to achieve Global Positioning System (GPS) records of 71,059 cars for research purpose without any personal information. This data covered 30 provinces in China, which could highly



improve our understanding on vehicle usage and to perform better in the aspects of comprehensiveness and accuracy.

In addition, several other new method and local data could be integrated to improve the inventory.
Previous studies usually calculated the provincial emissions using the local registration number, which was based on an assumption that all vehicles were running within the province or city where they registered. However, when it comes to freight trucks, this assumption is unwarranted. Instead of the traditional local registration based approach, we have developed a more-reasonable road emission intensity based (REIB) approach. Instead of relying on truck population from local registration database,
the spatial distribution of emission inventory is based on the total length of each road type in this province and also the emission intensity for this road type. Using this approach, NOx and PM emission for greatly improved for long-distance inter-province or inter-city cargo transportation (*Yang et al., 2015*).

The deficiencies in comprehensiveness and accuracy can also be improved by using local emission factors and speciation published recently (*Liu et al., 2009; Liu et al., 2015; Yao et al., 2015; Zhang et al., 2015;*
*Cao et al., 2016*). Instead of the emission factors given by common-used vehicle emission models developed by the U.S. and Europe, e.g., COPERT, MOVES, MOBILE and IVE, the measured local emission factors provided more reality to local emission level. Using chemical profiles obtained by experiments in western countries could not reflect the chemical characteristics of VOCs from vehicles in China accurately too. The recently speciation profiles were reported using China local fuel.

The national statistical data in China only provides the vehicle population classified by vehicle types (e.g., light-duty passenger vehicles, heavy-duty trucks). However, more-detailed vehicle population classified by fuel types and control technologies were required to calculate emissions because these two parameters have been acknowledged to influence emission factors distinctly (*Huo et al., 2012; Zhang et al., 2015; Cao et al., 2016*).

In this study, an updated speciation-based emission inventory of VOCs and an estimation of IVOCs from vehicles in China, with a target year of 2015, were developed using a set of state-of-the-art methods. The five deficiencies in comprehensive and accuracy mentioned above were solved one by one based on scientific calculating methodologies, big data and abundant local emission measurements. The IVOCs



emission factors were derived from US studies by matching vehicle emission categories between China
and US, because currently there were no local IVOCs emission factors reported.

## 2 Methodology and data

### 2.1 Vehicle stock and classification

In total, 22 types of on-road vehicles were considered in this study, including light-duty gasoline
passenger vehicles (LDGPVs), light-duty diesel passenger vehicles (LDDPVs), light-duty alternative-fuel
passenger vehicles (LDAPVs), medium-duty gasoline passenger vehicles (MDGPVs), medium-duty
diesel passenger vehicles (MDDPVs), medium-duty alternative-fuel passenger vehicles (MDAPVs),
heavy-duty gasoline passenger vehicles (HDGPVs), heavy-duty diesel passenger vehicles (HDDPVs),
heavy-duty alternative-fuel passenger vehicles (HDAPVs), gasoline taxis (GTAs), diesel taxis (DTAs),
alternative-fuel taxis (ATAs), gasoline buses (GBUs), diesel buses (DBUs), alternative-fuel buses (ABs),
light-duty gasoline trucks (LDGTs), light-duty diesel trucks (LDDTs), medium-duty gasoline trucks
(MDGTs), medium-duty diesel trucks (MDDTs), heavy-duty gasoline trucks (HDGTs), heavy-duty diesel
trucks (HDDTs) and gasoline motorcycles (GMs). For passenger vehicles, light-duty refers to the vehicle
whose length is less than 6000mm and ridership is less than or equal to 9. A medium-duty vehicle refers
to the length less than 6000mm and ridership among 10-19. A heavy-duty vehicle refers to the length
more than or equal to 6000mm or the ridership is equal to or more than 20. For trucks, a light-duty truck
refers to the length less than 6000mm and mass less than 4500kg. A medium-duty truck refers to the
length more than or equal to 6000mm or mass from 4500kg to 12000kg. A heavy-duty truck refers to the
truck whose mass is more than 12000kg. These vehicles were further classified by control technologies
(i.e., China 0, China 1, China 2, China 3, China 4 and above).

### 2.2 Vehicle activity

The real-world vehicle activity data used in this study was derived by statistical surveys, field tests and
literature review. To be specific, the detailed provincial population data of all types of vehicles excluding
GMs in 2015 was obtained by complete statistical survey conducted by the Vehicle Emission Control
Center (VECC) of China's Ministry of Environmental Protection (MEP), which could be regarded as



accurate as possible. The provincial GMs population in 2015 was obtained from China Automotive Industry Yearbook 2016.

The provincial annual vehicle kilometers traveled (VKT) data of light-duty passenger vehicles (LDPVs), which was the majority in the fleet and thus had the largest impact on the emission inventory, was acquired by processing and analysing the big data of GPS records (71059 cars). The driving frequency of different

types of trucks running on different kinds of roads (i.g., freeway, national road, provincial road and urban road) was acquired by analysing the survey data of 1060 valid questionnaires, which has been introduced in detail in our previous study (***Yang et al., 2015***). The annual VKT data for trucks were calculated based on report data from 2 million trucks in China.

In addition, for evaporative emission calculation, the provincial parking characteristics data, including

parking events numbers and parking durations, were also obtained by processing and analysing the GPS big data.

## 2.3 Vehicular emission data and estimation

The vehicular VOCs emissions at the provincial level were divided into three parts to be calculated,

including tailpipe emissions from non-truck vehicles (i.e., passenger vehicles, taxis, buses and motorcycles), tailpipe emissions from freight trucks and evaporation emissions from gasoline vehicles. Then the total provincial emission amounts in the year of 2015 were obtained by summing these three parts of emissions up. The emission factors for VOCs used here were derived by lab tests, field tests and literature review. The IVOCs emission calculation was very similar with VOCs, while only the tailpipe

exhaust is taken into consideration. For IVOCs emission factors, Zhao et al. reported a series measurements for gasoline and diesel vehicles (***Zhao et al.; 2016; Zhao et al.; 2015***) (Table S2 and Table S3). Details were introduced below.

### 2.3.1 Tailpipe emissions from non-truck vehicles

For a given province, the tailpipe VOCs and IVOCs emissions from non-truck vehicles were estimated

by Eq. (1):



$$E_{tailpipe,non-truck,i} = \sum_j \sum_k \left(EF_{tailpipe,j,k} \times VP_{i,j,k} \times VKT_{i,j}\right),$$

(1)

where $E_{tailpipe,non-truck,i}$ represents the annual tailpipe emissions from non-truck vehicles in province $i$ (g·year$^{-1}$); $EF_{tailpipe,j,k}$ represents the tailpipe VOCs/IVOCs emission factor of vehicle type $j$ with control technology $k$ (g·km$^{-1}$); $VP_{i,j,k}$ represents the registered population of vehicle type $j$ with control technology $k$ in province $i$; $VKT_{i,j}$ represents the annual VKT of vehicle type $j$ in province $i$ (km·year$^{-1}$).

For VTEs from all types of vehicles excluding trucks, the emission factors of VOCs obtained by abundant real-world emission tests conducted by our Tsinghua university research group and VECC of China's MEP were adopted (***Technical guidelines on emission inventory development of air pollutants from on-road vehicles (on trial)***) (Table S4).

For IVOCs emission factors, Zhao et al. reported a series measurements for gasoline and diesel vehicles (***Zhao et al.; 2016; Zhao et al.; 2015***). By considering vehicle model year, after-treatment devices and emission certification standard, each of the tested vehicles was matched to a category of China emission certification standard (Table S2). Thus, the emission factors for some vehicle categories were set up. For those categories without measurements (gasoline vehicles before China 1 and all diesel vehicles), emission factors were set as the same with China 1 category. For diesel passenger vehicles, the current IVOC emission factors were set as the same with the same level of gasoline vehicles. The IVOC emission factors were converted from the original unit of mg/kg-fuel to mg/km, using fuel economy. For each category, if there were more than one test available, the median of these emission factors would be used as the emission factor of this type of vehicle. The detailed emission factors were listed in Table S5.

### 2.3.2 Tailpipe emissions from freight trucks

Considering the fact that a majority of freight trucks are used for long-distance inter-city or inter-province cargo transportation, REIB approach instead of the traditional local registration based approach was utilized to calculate truck emissions, as was detailly described in our previous work (***Yang et al., 2015***). The provincial tailpipe emissions from freight trucks were estimated by Eq. (2):

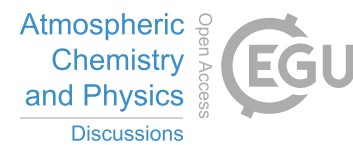

$$E_{tailpipe,truck,i} = \sum_j \sum_k \sum_m \left[ \frac{(EF_{tailpipe,j,k} \times VP_{j,k} \times VKT_{j,k} \times DP_{j,m}) \times L_{i,m}}{L_m} \right],$$

(2)

where $E_{tailpipe,truck,i}$ represents the annual tailpipe VOCs/IVOCs emissions from freight trucks in province $i$ (g·year$^{-1}$); $EF_{tailpipe,j,k}$ represents the tailpipe VOCs/IVOCs emission factor of vehicle type $j$ with control technology $k$ (g·km$^{-1}$); $VP_{j,k}$ represents the national population of vehicle type $j$ with control technology $k$; $VKT_{j,k}$ represents the annual VKT of vehicle type $j$ with control technology $k$ (km·year$^{-1}$); $DP_{j,m}$ represents the distance portion of vehicle type $j$ running on road type $m$; $L_{i,m}$ and $L_m$ represents the total length of road type $m$ in province $i$ and in China, respectively (km).

For VTEs from trucks, we used the operating-mode-bin-based method introduced in our previous study to investigate the real-world emission factors for VOCs (*Yang et al., 2015*). First, the second-by-second vehicle-specific power (VSP) and engine stress (ES) were calculated using the GPS records of 16 trucks and the equations suggested by MOVES model and IVE model, respectively. Then thirty operating mode bins were identified based on the VSP and ES data and the time fraction of each bin was given. Finally, the distance-based emission factors for different types of trucks running on different kinds of roads were calculated according to the emission rate of each bin, which was based on our previous test results (*Liu et al, EST, 2009*).

For IVOCs emission factors, a mapping to match US emission certification level to China emission level was built (Table S3). Then the emission factors for US fleet were converted to emission factors for China trucks. There was no data available for most categories. We have to make assumptions to fill the gap: (1) Mini and light-duty trucks for the same control level shared the same emission factors. (2) Medium and heavy-duty trucks had same emission factors which was 50% higher than light-duty trucks. This ratio was similar to the VOCs or POA emission ratios of heavy/light for trucks. (3) For those emission levels without data, the emission factors for neighbour emission level would be used. The final assumption for IVOC emission factors were introduced in Table S5.



### 2.3.3 Evaporation emissions from gasoline vehicles- Diurnal and hot soak

Hot soak and diurnal emissions both occur when vehicles are parked. Diurnal loss is defined as the gasoline vapors that are generated and emitted while vehicles are parked. The emission factors of diurnal and hot soak were obtained by a set of Sealed Housing for Evaporative Determination (SHED) tests, as was introduced in our previous study (**Liu et al., 2015**). The provincial annual diurnal emissions from non-GM gasoline vehicles and GMs were calculated by Eq. (3)-(6) and Eq. (7) respectively.

$$E_{diur,non-GM,i} = E_{diur,<24,non-GM,i} + E_{diur,24-48,non-GM,i} + E_{diur,>48,non-GM,i} \, ,$$

$$(3)$$

$$E_{diur,<24,non-GM,i} = \left[ EF_{diur,<24,LDGPVs} \times \left( P_{duration,1-24,i} \times T_i - P_{event,1-24,i} \times N_i \times 1 \right) \right] \times 365 \times$$
$$P_{i,non-GM} \, ,$$

$$(4)$$

$$E_{diur,24-48,non-GM,i} = \left[ EF_{diur,<24,LDGPVs} \times P_{event,24-48,i} \times N_i \times 23 + EF_{diur,24-48,LDGPVs} \times \right.$$
$$\left. \left( P_{duration,24-48,i} \times T_i - P_{event,24-48,i} \times N_i \times 24 \right) \right] \times 365 \times P_{i,non-GM} \, ,$$

$$(5)$$

$$E_{diur,>48,non-GM,i} = \left[ EF_{diur,<24,LDGPVs} \times P_{event,>48,i} \times N_i \times 23 + EF_{diur,24-48,LDGPVs} \times P_{event,>48,i} \times \right.$$
$$N_i \times 24 + EF_{diur,48-72,LDGPVs} \times \left( P_{duration,>48,i} \times T_i - P_{event,>48,i} \times N_i \times 48 \right) \right] \times$$
$$365 \times P_{i,non-GM} \, ,$$

$$(6)$$

where $E_{diur,non-GM,i}$ represents the total annual diurnal (simultaneous permeation included) emissions from non-GM gasoline vehicles registered in province $i$ (g·year$^{-1}$); $E_{diur,<24,non-GM,i}$, $E_{diur,24-48,non-GM,i}$, $E_{diur,>48,non-GM,i}$ represents the annual diurnal (simultaneous permeation included) emissions occurred respectively in the first day, second day and third day and after of parking (g·year$^{-1}$); $EF_{diur,<24,LDGPVs}$, $EF_{diur,24-48,LDGPVs}$, $EF_{diur,48-72,LDGPVs}$ represents the measured diurnal (simultaneous permeation included) emission factors of China 4 LDGVs (g·hour$^{-1}$); $P_{event,1-24,i}$, $P_{event,24-48,i}$, $P_{event,>48,i}$ represents the percentage of parking events that are 1-24 hours, 24-48 hours



and above 48 hours; $P_{duration,1-24,i}$, $P_{duration,24-48,i}$, $P_{duration,>48,i}$ represents the percentage of total

parking duration that are between 1 hour and 24 hours, between 24 hours and 48 hours and above 48

hours.

$$E_{diur,GMs,i} = EF_{diur,GMs} \times VP_{i,GMs} \times VKT_{i,GMs} \ ,$$

(7)

where $E_{diur,GMs,i}$ represents the annual diurnal (simultaneous permeation included) emissions from GMs

registered in province $i$ (g·year$^{-1}$); $EF_{diur,GMs}$ represents the diurnal (simultaneous permeation included)

emission factor of GMs (g·hour$^{-1}$).

According to the US EPA, hot soak is defined as the evaporative losses that occur within the one-hour

period after the engine is shut down (**EPA420-R-01-026**). If the parking duration is longer than one hour,

then the extra vapor losses fall into diurnal emissions. The provincial hot soak emissions for non-GM

gasoline vehicles (i.e., LDGPVs, MDGPVs, HDGPVs, GTs, GBUs, LDGTs, MDGTs, HDGTs) and GMs

were calculated by Eq. (8) and Eq. (9) respectively:

$$E_{soak,non-GM,i} = EF_{soak,LDGPVs} \times \left[ \left( T_i \times 365 \times P_{duration,<1,i} \right) + \left( N_i \times 365 \times P_{event,>1,i} \times 1 \right) \right] \times$$
$$VP_{i,non-GMs} \ ,$$

(8)

where $E_{soak,non-GM,i}$ represents the annual hot soak (simultaneous permeation included) emissions from

non-GM vehicles in province $i$ (g·year$^{-1}$); $EF_{soak,LDGPVs}$ represents the hot soak (simultaneous permeation

included) emission factor of LDPGVs (g·hour$^{-1}$); $T_i$ represents the annual average parking duration per

day per vehicle of province $i$ (hour); $N_i$ represents the annual average parking events per day per vehicle

of province $i$; $P_{duration,<1,i}$ represents of the percentage of total parking duration shorter than 1 hour of

province $i$; $P_{event,>1,i}$ represents of the percentage of parking events with a duration shorter than 1 hour

of province $i$; $VP_{i,non-GMs}$ represents the non-GM gasoline vehicle population of province $i$.

$$E_{soak,GMs,i} = EF_{soak,GMs} \times VP_{i,GMs} \times VKT_{i,GMs} \ ,$$

(9)




where $E_{soak,GMs,i}$ represents the annual hot soak (simultaneous permeation included) emissions from GMs registered in province $i$ (g·year⁻¹); $EF_{soak,GMs}$ represents the hot soak (simultaneous permeation included) emission factor of GMs (g·hour⁻¹); For VEEs from GMs, the emission factors given by the International Council on Clean Transportation (ICCT) were utilized (***ICCT, 2012***). $VP_{i,GMs}$ represents the GMs population registered in province $i$; $VKT_{i,GMs}$ represents the annual VKT of GMs in province $i$ (km·year⁻¹).

**2.3.4 Evaporation emissions from gasoline vehicles- Refuelling**

China is following European control experiences to popularize Stage-II vapor control system in refuelling stations to reduce refuelling loss. The vehicle refuelling emissions were also measured by our team from SHED tests (***Yang et al, 2015***). The provincial refuelling emissions from gasoline vehicles were calculated by Eq. (10):

$$E_{refuel,i} = EF_{refuel}\times[(1-\theta)\times w_i + (1-w_i)]\times CF_i ,$$

$$(10)$$

where $E_{refuel,i}$ represents the annual refuelling emissions from gasoline vehicles in province $i$ (g·year⁻¹); $EF_{refuel}$ represents the refuelling emission factor (g·L⁻¹); $\theta$ represents the average efficiency of the Stage-II vapor control system, 82% in this study according to our measurements in Beijing; $\omega_i$ represents the percentage of filling stations equipped with Stage-II vapor control system in province $i$, 100% in Beijing, 90% in Shanghai and Guangdong, 60% in Tianjin and Hebei and 0 in other provinces in this study according to survey; $CF_i$ represents the annual motor gasoline consumption of province $i$ (L·year⁻¹).

**2.3.5 Evaporation emissions from gasoline vehicles- Running loss**

The vehicle running loss happens when the engine is on. However, this emission is not from tailpipe, but from the fuel system. The provincial annual running loss emissions from non-GM gasoline vehicles and GMs were calculated by Eq. (11) and Eq. (12) respectively:





$$E_{running,non-GM,i} = EF_{running,LDGPVs}\times(24 - T_i)\times365\times VP_{i,non-GM} ,$$

$$(11)$$

where $E_{running,non-GM,i}$ represents the annual running loss emissions from non-GM gasoline vehicles registered in province $i$ (g·year$^{-1}$); $EF_{running,LDGPVs}$ represents the running loss emission factor of LDGVs (g·L$^{-1}$). The emission factors of running loss were acquired from MOVES model due to the lack of local lab test results (***MOVES, 2010***).

$$E_{running,GMs,i} = EF_{diur,GMs}\times VP_{i,GMs}\times VKT_{i,GMs} ,$$

$$(12)$$

where $E_{running,GMs,i}$ represents the annual running loss (simultaneous permeation included) emissions from GMs registered in province $i$ (g·year$^{-1}$); $EF_{running,GMs}$ represents the running loss (simultaneous permeation included) emission factor of GMs (g·hour$^{-1}$). For VEEs from GMs, the emission factors given by the International Council on Clean Transportation (ICCT) were utilized (***ICCT, 2012***).

**2.4 Species analysis**

The vehicular VOCs emissions speciation was further determined by Eq. (13):

$$E_{speciated} = E_{tailpipe,gasoline}\times PR_{tailpipe,gasoline} + E_{tailpipe,diesel}\times PR_{tailpipe,diesel} + E_{evap}\times$$
$$PR_{evap} ,$$

$$(13)$$

where $E_{speciated}$ represents the speciated annual VOCs emissions from on-road vehicles registered in province $i$ (g·year$^{-1}$); $E_{tailpipe,gasoline}$, $E_{tailpipe,diesel}$, and $E_{evap}$ represents the annual tailpipe VOCs emissions from gasoline vehicles (alternative-fuel vehicles included), the annual tailpipe VOCs emissions from diesel vehicles and the annual evaporative VOCs emissions respectively; $PR_{tailpipe,gasoline}$, 310 $PR_{tailpipe,diesel}$, and $PR_{evap}$ represents the measured VOCs profiles of tailpipe emissions from gasoline vehicles, tailpipe emissions from diesel vehicles and evaporative emissions, respectively.



The VOCs profiles used in this study to generate the speciated vehicular VOCs emission inventory were derived from literature review and lab tests. For tailpipe VOCs emissions from gasoline vehicles and diesel vehicles, the corresponding local profiles were reported by Yao et al. according to on-board exhaust tests with 18 in-use diesel trucks and 30 in-use light-duty gasoline vehicles in Beijing (***Yao et al, 2015*** and ***Cao et al, 2016***). For exhaust emissions, the profiles [1, 2]. For vehicle evaporative emissions, a comprehensive species profile was obtained based on results from the 30 cross-over evaporative tests we conducted before (***Man et al., 2016***).

## 3 Results and discussion

### 3.1 Activity characteristics of vehicles

#### 3.1.1 Vehicle population

In the year of 2015, the total population of the 22 types of on-road vehicles in China was 259 million, to which GMs and non-GM vehicles contributed 34.3% and 65.7%, respectively (Figure 1). Among the non-GM vehicles, LDPVs were the predominant contributor, with a proportion of 81.0%, followed by light-duty trucks (LDTs, 9.4%), heavy-duty trucks (HDTs, 3.6%), taxis (TAs, 2.3%), medium-duty trucks (MDTs, 1.7%), medium-duty passenger vehicles (MDPVs, 0.8%), heavy-duty passenger vehicles (HDPVs, 0.7%) and buses (BUs, 0.5%). In terms of control technologies, China 3 vehicles accounted for the largest proportion (51.0%) in China's non-GM vehicle fleet, followed by China 4 (22.9%), China 1 (9.5%), China 2 (7.2%), China 5 vehicles (5.2%) and China 0 (4.2%). This fleet structure varies when new vehicles add into the fleet. For example, in 2012, the China 2 still occupies 15.69% in total, while China 4 only has 10.12%. Similarly, percent for China 1 was reduced from 14.92% in 2012 to 9.5% in 2015. In terms of automotive fuels, gasoline was the most widely-used fuel of non-GM vehicles in China, with a proportion of 86.0%. Comparatively, diesel and alternative fuels were consumed much less, with a respective proportion of 12.9% and 1.2%. Table 1 lists the vehicle population classified by fuel types. LDPVs, MDPVs and TAs were mainly fuelled by gasoline while HDPVs, LDTs, MDTs, HDTs and BUs were primarily fuelled by diesel. For both passenger vehicles and freight trucks, the heavier the vehicles were, the larger the percentage of vehicles using diesel as fuels became. Alternative fuels were still not a mainstream option, especially in the freight truck fleet. However, the percentages of BUs and TAs using



alternative fuel both exceeded 8.7%, which were mainly resulted from the strong promotion of the central

and local governments.

### 3.1.2 VKT characteristics of LDPVs

In total, GPS records of 71,059 cars running in 30 provinces from July 1, 2014 to July 1, 2015, including 931,581,667 km driving distances and 1,585,771,787,511 valid seconds, were collected and analysed to obtain the real-world VKT characteristics of LDPVs in different provinces. It was found that the national

average VKT of LDPVs in China was 18,886±10,469 km per vehicle per year. Table 2 gives the provincial annual average VKT values with vehicle sample sizes, while Figure 2 shows the distribution characteristics of annual VKT data in each province. The annual average VKT of LDPVs in Beijing and Shanghai, which were both among the highest developed cities in the world, were much lower than the national average value given by this study. And on the time scale, they were much lower than the

corresponding local values given by surveys conducted ten years ago in 2004 (***Liu et al., 2007***). This phenomenon was mainly caused by three reasons. Firstly, the per capita ownership of cars in Beijing and Shanghai during our sampling periods was much higher than the national average value around the same time and the corresponding local values ten years ago. A certain amount of families in these two cities own more than one car nowadays, leading to the decrease of annual VKT of each car under the

circumstances that their regular commuting distances have not changed that much. Secondly, the most stringent traffic control policies ever have been implemented in Beijing and Shanghai in the past several years, resulting in longer parking duration and smaller annual VKT of vehicles in these two cities. Thirdly, a number of citizens in Beijing and Shanghai have chosen to increase the percentage of traveling by public transportation facing the growing traffic jams in peak hours.

### 3.1.2 Vehicle parking characteristics

Totally, GPS records including 102,576,888 continuous parking events and 11,465,694,907,444 valid seconds, were collected and analyzed to get the real-world vehicle parking characteristics in China, which have significant impacts on VEEs. It was found that the parking characteristics in all the provinces except for Beijing were quite similar. For vehicles in Beijing, the annual average number of parking events per





day per vehicle was 3.89 while the annual average parking duration per day per vehicle was 22.21 h. For vehicles in the other provinces, however, these two values were 5.73 and 22.11 h, respectively.

Figure 3(a) and Figure 3(b) shows the distribution of parking events and total parking duration in six time intervals (0-1 h, 1-24 h, 24-48 h, 48-72 h, 72-119.5 h, >119.5 h) in Beijing and other provinces, respectively. Parking events falling into the first two time intervals (<24 h), though having percentages of 95.5-98.8% in number, only accounted for 51.3-76.8% of the total parking duration, showing that the current VEEs control policy in China, where only VOCs evaporated in the first 24 h of parking is given a limitation value, cannot effectively cover the majority of VEEs in China. The next phase of emission control, China 6 emission standard, will enhance the evaporative emission control by adding 48 h duration into consideration. Overall, Beijing was found to have fewer parking events but higher percentages in parking events with duration longer than 1 h, thus resulting in longer total parking duration. This phenomenon was mainly caused by the consistent traffic control measures implemented in Beijing dating from the 2008 Beijing Olympic Games, where the days in which vehicles can be driven within the Fifth Ring Road is strictly limited.

## 3.2 Emissions

### 3.2.1 Vehicle VOCs emissions

In the year of 2015, China's on-road vehicles emitted 4.21 Tg VOCs totally (Figure 4). Figure 4a-f shows the provincial results of vehicular VOCs emissions in 2015. VTEs were still the predominant contributor to the total VOCs emission amount (Figure 4a), with a proportion of 60.80%. However, VEEs, which were responsible for the other 39.20% of emissions, should also be paid great attention to. For the provinces that own large fleet of light-duty vehicles, eg. Guangdong, Shandong and Jiangsu were ranked as the top on the league table, with a respective contribution of 10.66%, 8.85% and 6.54% to the total amounts of VOCs from vehicles. There were slightly difference between the ranking of tailpipe exhaust and evaporation (Figure 4b and 4c), which was because of the difference on vehicle fleet, vehicle parking behaviour and ambient temperature. Figure 4b provided insight for evaporative emissions for gasoline vehicles except motorcycles, although the evaporation from motorcycles were included in Figure 4a. Refuelling was important (7.83%), but it could be controlled immediately by stage-II system in service



stations. Thus, the major challenge for the future is from running loss (81.05%) and diurnal (10.00%). The new China 6 vehicle emission standard will help to control diurnal emissions, which will target at one of the major evaporative sources. However, running loss is still a big issue. The current estimation

showed it could be the most important part for vehicle evaporation for now and especially for future. Since we don't have any measurements data in China for running loss, uncertainties for this calculation is the largest.

Figure 4c-4f provided sub-classification for tailpipe exhausts only. When controlling tailpipe VOCs, passenger vehicles, trucks and motorcycles should all be considered. Their contributions are 49.86%,

28.15% and 21.99% separately. We have paid a lot of attention to passenger cars, but maybe too less attention to motorcycles and trucks. For public transportation, taxis and buses contributed 12.22% and 2.04% of total tailpipe emissions, while the population for these categories are only 2.3% and 0.5%. Motorcycles were still important sources in Guangdong, Shandong, Yunnan, Hunan, Guangxi and Fujian, in where the motorcycles markets were still large. LDPVs and LDTs were two dominant subcategories

for VOCs emissions. In this study, the activity data for LDPVs were greatly improved to reduce the uncertainty. If possible, future studies were expected to get more detailed information for LDTs. China 0 vehicles still contributed most (35.19%) to the tailpipe VOCs. Vehicles before China 4 (not included) contributed 94.67% of the total tailpipe emissions.

### 3.2.2 Vehicle IVOCs emissions

In the year of 2015, China's on-road vehicles emitted 121.23 Gg IVOCs totally. Figure 5 was the IVOCs emissions by provinces. It should be noted here, the estimation was totally based on emission factors from the tests in United States. The current estimation can only provide insight of the order of magnitude compared with VOC emission level based on the same input of vehicle activity data. Diesel vehicles contributed 58.31% to the IVOCs, which is a little bit higher than gasoline vehicles. Passenger vehicles

and trucks contributed half to half.



### 3.2.3 Speciated vehicular VOCs emission inventory

In total, 97 species of evaporation, 30 species of gasoline exhaust and 20 species of diesel exhaust were recognized (Figure 6). The detailed emission amounts were provided for 36 species, while the others are listed in category named "others". Toluene, i-pentane, benzene are main species from gasoline exhaust.

N-butane, i-pentane, i-butane and propane are main species from evaporation. Evaporation shared 83.66% vehicle emissions of n-butane, i-pentane, and i-butane. Formaldehyde and acealdehyde were most contributed by diesel tailpipe emissions. Our speciation profile for the evaporative emission has very little percentage of unresolved complex mixture (UCM). In the future, the speciation profile for exhaust still need to be improved for both gasoline and diesel vehicles.

### 3.2.4 Implication to emission control

Figure 7 compared total emissions, emission intensity by area and emission per vehicles among provinces. It's not a surprise that the highest emission intensity by area happened in most developed regions, eg. Beijing, Tianjin and Shanghai (4500~9000 kg km$^{-2}$). In the other provinces, the emission intensity varies in 13~2700 kg km$^{-2}$. So, if we only provide league table by total emissions, the readers may be misleading

to avoid the importance of emission control in such areas. Beijing has the lowest emissions per vehicle compared with other regions (14~85 kg per vehicle), which mean the vehicles were the cleanest in Beijing. Thus, continuing reducing emissions from these highly-developed regions would be quite difficult from only technology aspect. The extra efforts from reducing population intensity, to change human behaviour on using vehicles would be considered as the main strategies for these regions.

### 3.3 Uncertainty analysis

Inevitable uncertainties come into the VOCs emission inventories due to the use of different input data, including activity characteristics, emission factors and VOCs emission profiles. Figure 8 compared this updated emission inventory with previous studies. Our result was in the same order of magnitude of previous estimation and higher than Wei's forecast in 2011 (***Wei et al, 2011***). Following reasons may

explain the differences. For the first time, vehicle evaporative emission was taken into consideration in detail, which added on the total VOCs. Secondly, the vehicle usage data was derived from big data which



were survey for those 'live' vehicles. The VKT used in our estimation is based on vehicle age. So the emissions from old vehicles were reduced. To further reduce the uncertainty of VOCs emission inventory, activity for LDTs, China 0 and China 1 vehicles should be improved. These categories now still
contributed significantly in total emission amount. However, due to author's experience, the usage data of these vehicles had the largest uncertainty among all categories.

It's very hard to evaluate the uncertainty of IVOC emission inventory. The major uncertainty was from the emission factors. IVOC measurement is getting popular in all around the world recently. This will help to build the emission factor database and then reduce the uncertainty greatly.

The uncertainty for species profile was significant for exhausts and negligible for evaporation. The profile used for evaporation was a comprehensive profile combining vehicle activity, technology contribution in fleet and profiles for different processes. Thus, the profile for evaporation was representative. In addition, the species of evaporation almost reached 100% of total hydrocarbons, which providing enough resolution for specie profile. All the reasons above successfully reduced the uncertainty of the species based
inventory for evaporation. The uncertainty for exhaust species were mainly from three aspects. Firstly, the current profile was based on individual test results and no comprehensive profile was built to represent the fleet average. Secondly, even for the individual tests, the recognized species was too less from the VOCs analysis. Thirdly, the species formed from incomplete combustion and from unburned fuel were not understood very well, which brought difficulties on build species profiles for exhaust.

**4 Conclusion**

The advantages of this study included: updated vehicle activity data from more than 70,000 cars and trucks in 30 provinces, detailed vehicle fleet statistic, the first-hand evaporation data and REIB framework to account inter-provinces transportation for trucks. The total VOCs emissions from on-road vehicles in China were about 4.21 Tg in 2015.

To improve the emission inventory, the emission factors for running loss of evaporation are urgently needed. The IVOC emission factors for all kinds of vehicles are urgently needed. The activity data for LDTs and old vehicles should be improved. The species profiles for exhaust, especially for gasoline vehicles are still weak.



To providing insights on vehicle emission controls, we suggest to pay more attention to reduce the

population density and vehicle usage in highly-developed regions. These measures should be a package

policy to deal with both the traffic congestion and also emissions.

*Acknowledgements*. This work is supported by the National Natural Science Foundation of China (41571447), the Major Research plan of the National Natural Science Foundation of China (Grant No.91544110), the National Program on Key Basic
Research Project (2014CB441301). H. Liu was supported by the National Natural Science Foundation of China (71101078). We would like to thank Hao Xu for suggestions on figure design.

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



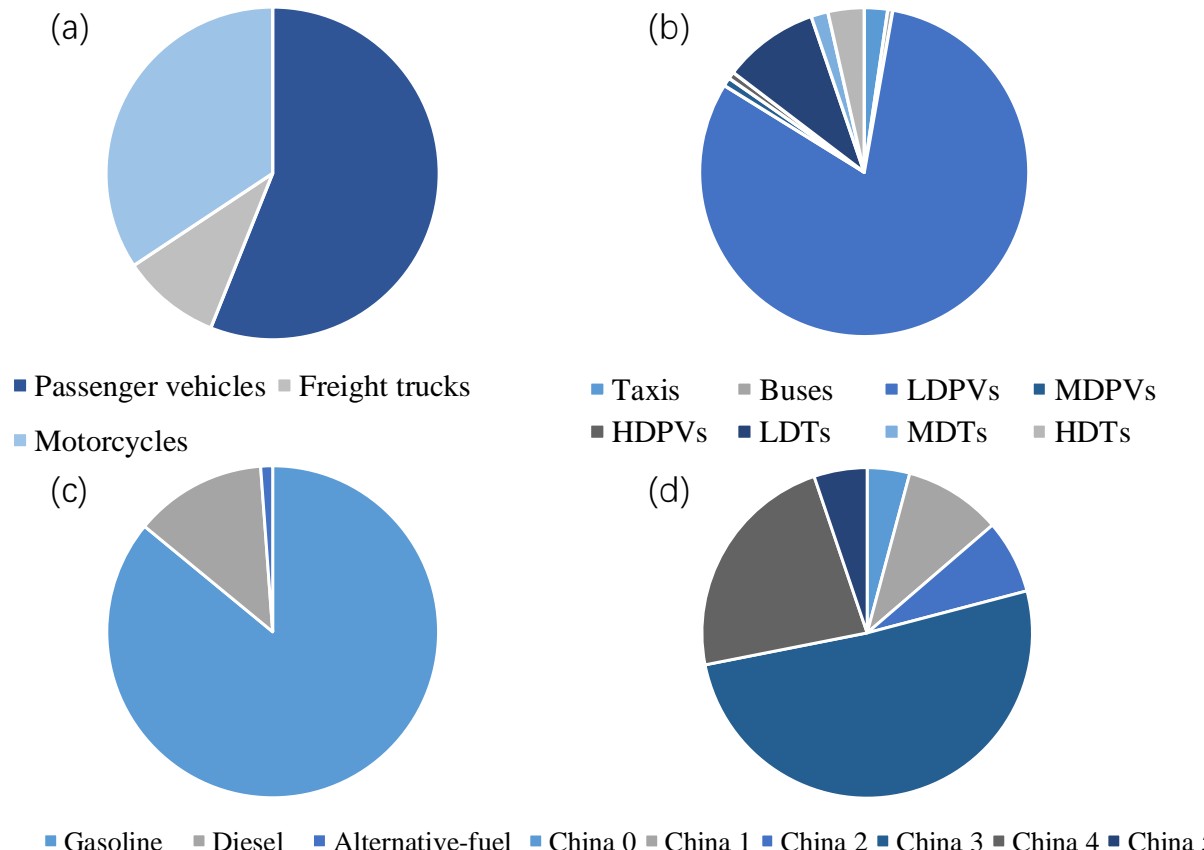

**Figure 1. Percentage of vehicle ownerships.**





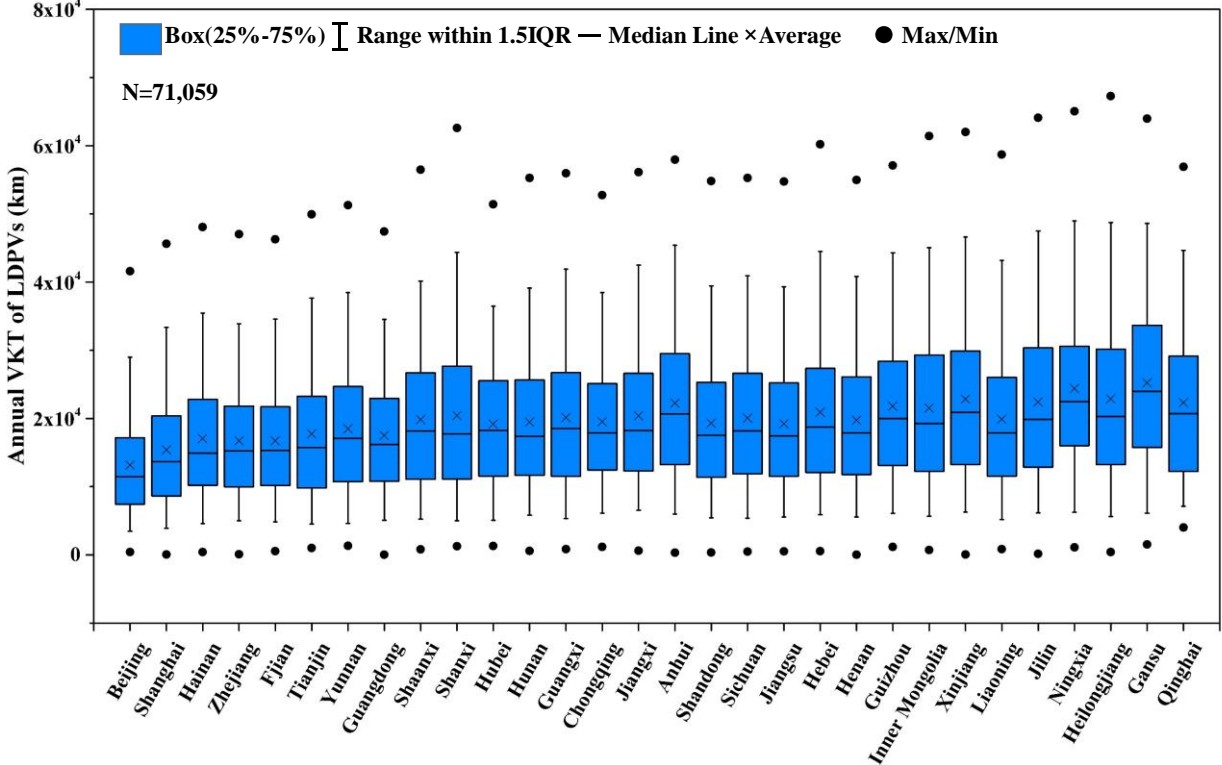

**Figure 2. Provincial annual VKT of LDPVs in China.**



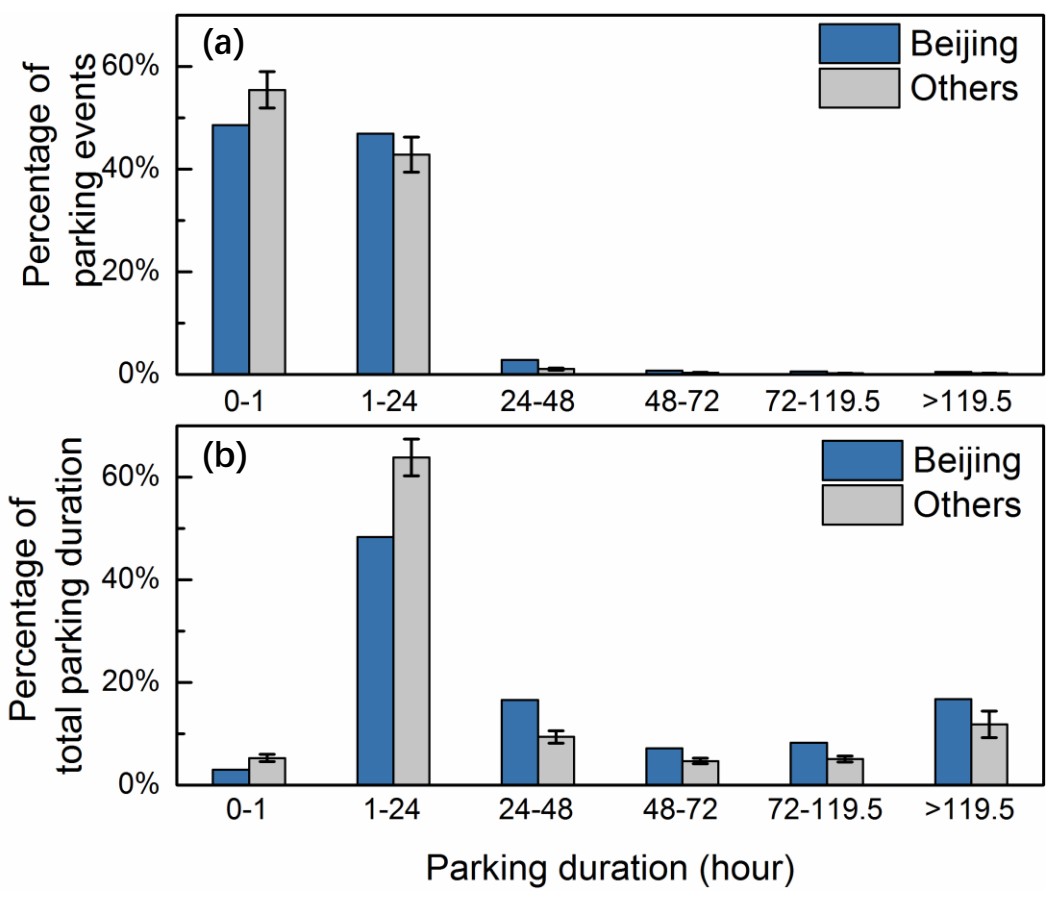


**Figure 3. Real-world parking duration distribution (a) percentage of parking events, (b) percentage of total parking duration.**





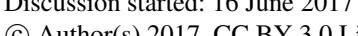

**Figure 4. Provincial VOCs emissions from vehicles in 2015 (a) total emission amount classified by emission sources, (b) evaporation emission amount classified by evaporation processes (motorcycles excluded), (c) tailpipe emission amount classified by vehicle types, (d-f) tailpipe emission amounts classified by detailed categories, emission certification levels and fuel type (motorcycles excluded).**





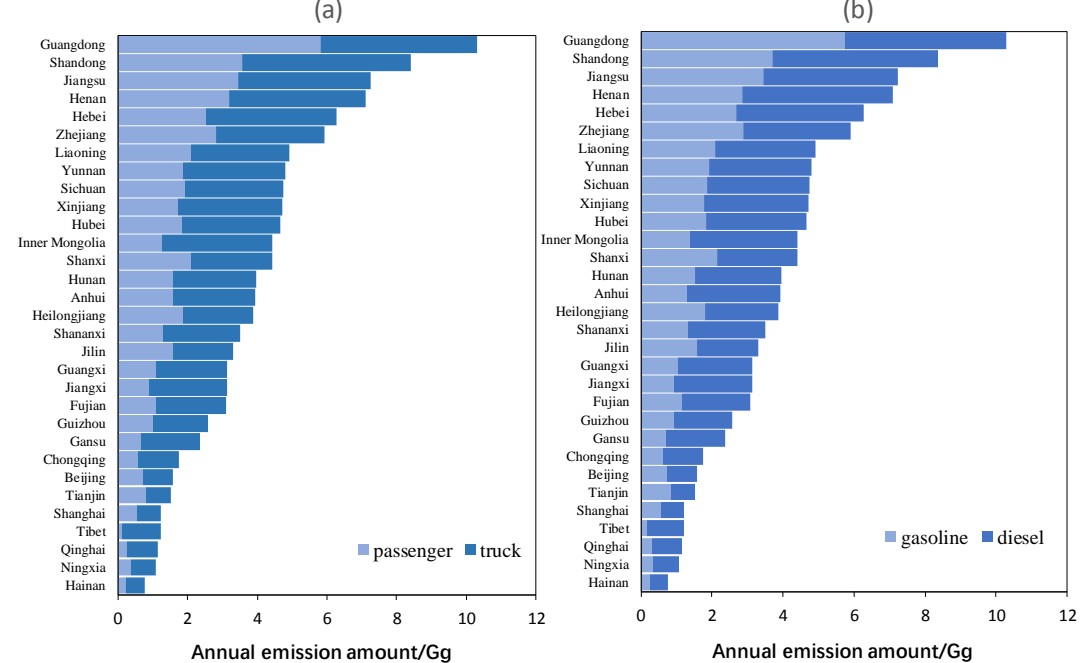


**Figure 5. Provincial IVOCs emissions from vehicles in 2015 (a) total emission amount classified by vehicle types, (b) total emission amount classified by emission sources.**





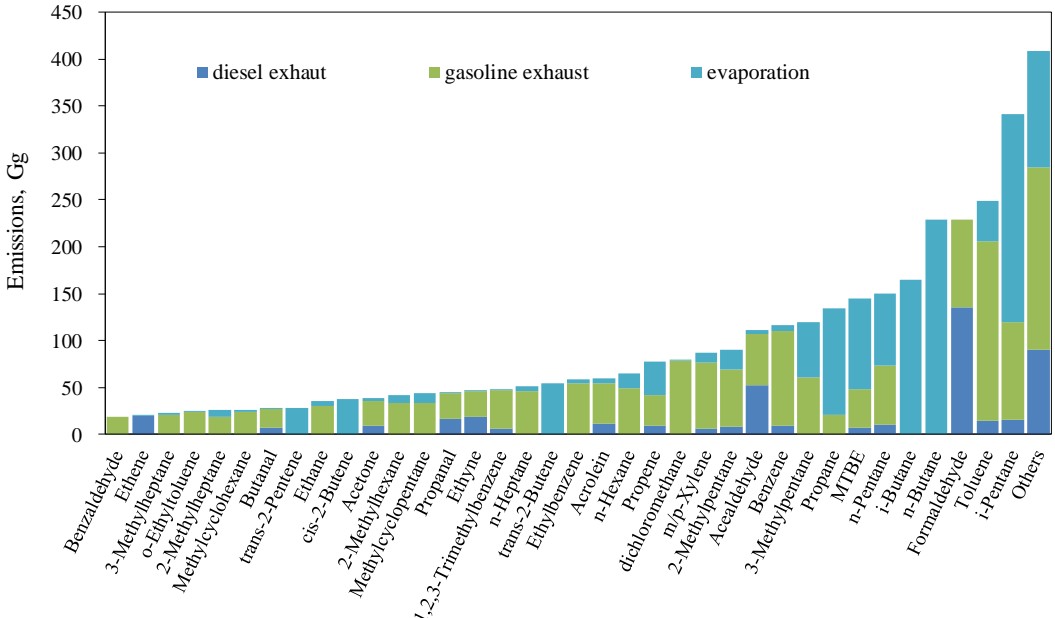


**Figure 6. Speciated VOCs components emissions classified by emission source**





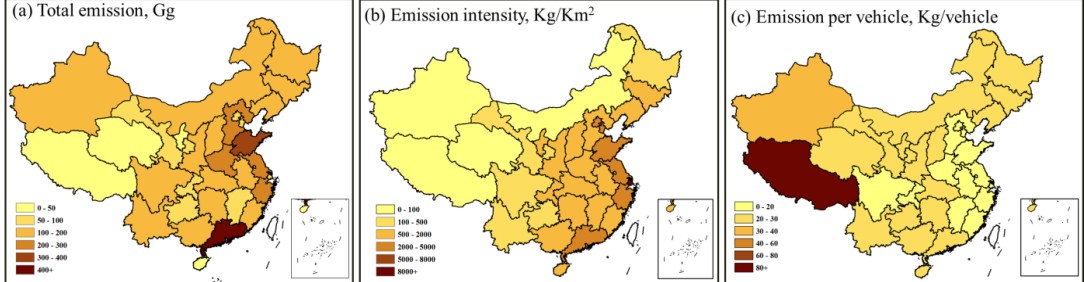

**Figure 7. Province based emission analysis (a) total emission amount, (b) emission intensity, (c) emission per vehicles (motorcycles were excluded).**





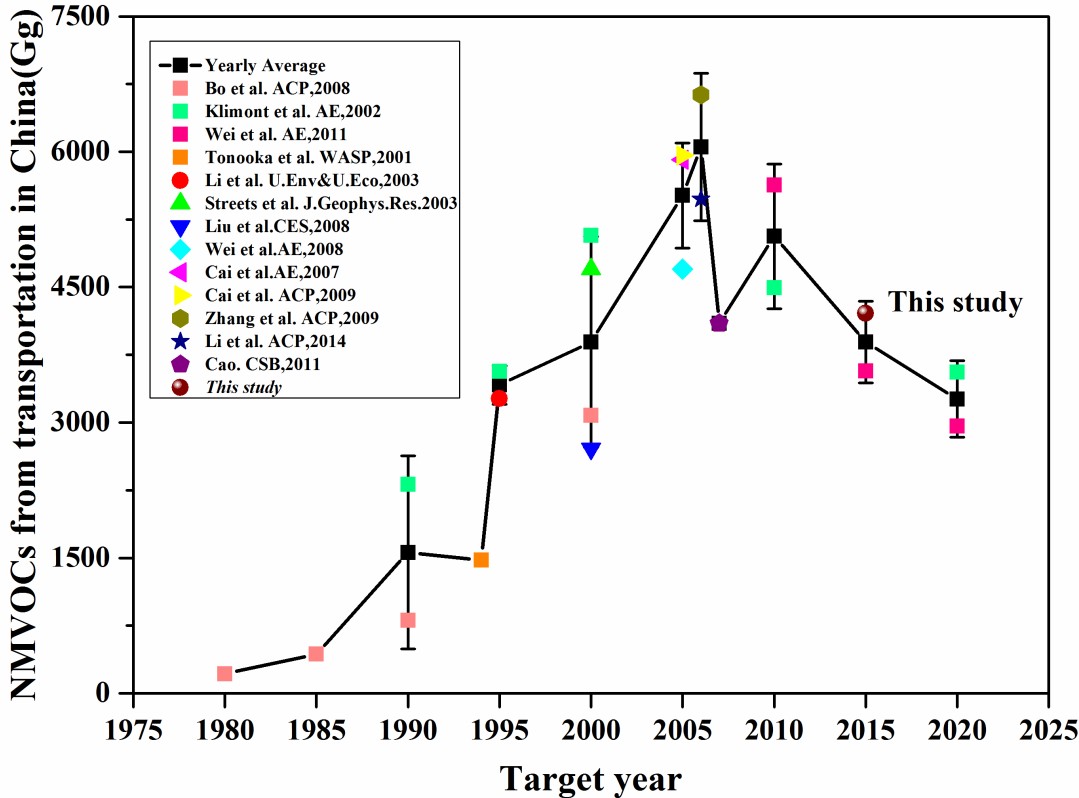

**Figure 8. Comparing this study to previous transportation emission inventories.**





**Table 1 Population of different types of vehicles in China in the year of 2015**

| Vehicle type | Population percentage (%) | | |
|---|---|---|---|
| | Gasoline | Diesel | Alternative fuels |
| LDPVs | 97.96 | 1.15 | 0.90 |
| MDPVs | 56.53 | 40.68 | 2.78 |
| HDPVs | 15.97 | 75.03 | 9.00 |
| LDTs | 41.50 | 58.50 | 0.00 |
| MDTs | 18.92 | 81.08 | 0.00 |
| HDTs | 7.65 | 92.35 | 0.00 |
| TAs | 61.89 | 29.37 | 8.74 |
| BUs | 13.76 | 55.39 | 30.85 |



**Table 2 Provincial annual average VKT of LDPVs in China**

| Province | Vehicle sample size | Annual average VKT (km) |
|---|---|---|
| Beijing | 2645 | 13169±7741 |
| Shanghai | 3833 | 15389±8972 |
| Hainan | 581 | 16941±9508 |
| Zhejiang | 6356 | 16740±8897 |
| Fujian | 3059 | 16726±8784 |
| Tianjin | 772 | 17785±10308 |
| Yunnan | 1370 | 18609±10307 |
| Guangdong | 16553 | 17503±8952 |
| Shaanxi | 1766 | 19866±10964 |
| Shanxi | 1225 | 20466±12131 |
| Hubei | 976 | 19313±9669 |
| Hunan | 1320 | 19524±10545 |
| Guangxi | 1086 | 20251±11231 |
| Chongqing | 1279 | 19529±10022 |
| Jiangxi | 903 | 20406±10982 |
| Anhui | 1007 | 22209±11744 |
| Shandong | 2449 | 19333±10420 |
| Sichuan | 1984 | 20120±10959 |
| Jiangsu | 5066 | 19238±10331 |
| Hebei | 2933 | 20915±11594 |
| Henan | 1818 | 19759±10693 |
| Guizhou | 746 | 21985±11800 |
| Inner Mongolia | 2322 | 21660±12118 |
| Xinjiang | 991 | 22901±12122 |
| Liaoning | 4049 | 19953±11365 |
| Jilin | 1386 | 22400±12630 |
| Ningxia | 418 | 24345±12810 |
| Qianghai | 171 | 22488±12265 |
| Heilongjiang | 1552 | 23008±13102 |
| Gansu | 443 | 25460±12659 |

[a] There is no VKT data for Tibet and we used the national average, which was calculated using the data of the other 30 provinces, to represent the annual VKT of Tibet in this study.