# Peer review of "An updated emission inventory of vehicular VOCs/IVOCs in China"

_Atmospheric Chemistry and Physics, 2017_

## Referee Comment (RC1) · Anonymous Referee #1 · 12 Aug 2017

In this work, Liu et al. developed an updated VOC emission inventory for the on-road vehicles in China for 2015. Particularly, they refined their analysis by using vehicle activity data collected from a large number of GPS records, taking into account VOC evaporation emissions from gasoline vehicles, and including tailpipe IVOC emission estimates. The topic is suitable for the Atmospheric Chemistry and Physics and the technical part of the manuscript is relatively well described. However, there are some technical, editorial, and grammatical issues in the current manuscript that need to be clarified and corrected first. The manuscript is highly suggested to be grammar-checked by native English speakers. Considering the results of this work is potentially of great value to the atmospheric modeling community, the reviewer suggests a carefully revised manuscript for publication in ACP.

[Figure]

Abstract line 27. The authors state in the abstract that the VKT level of "trucks were calculated from reported data by more than 2 million trucks in China". According to the Chinese official statistics, there were ~20 million trucks in China in 2015. That means the authors have collected VKT data of ~10%, which is a quit decent sampling ratio, of all trucks in China. However, relevant results are neither described nor referenced to previous studies in the main text (e.g., Sect. 2.2 and 3.1.2). The reviewer suggests the authors adding the description and discussion paragraphs or sections to introduce the methods and results in detail.

The section heading of Sect. 2.1 is "Vehicle stock and classification". However, only vehicle classification is described.

The classification of vehicles is not very clear to the reviewer and needs more clarification. First, the criterions to distinguish LD, MD, and HD passenger vehicles and to distinguish LD, MD, and HD trucks are not given. Second, if taxis are classified as separate vehicle types, the authors should add a statement previously that LD, MD, and HD passenger vehicles do not include taxis. Third, the reviewer is wondering what kind of vehicles are treated as alternative-fuel vehicles? Electric? Plug-in electric? Hybrid? Internal combustion engine vehicles running on alternative fuels such as CNG/LNG/LPG, methanol, or ethanol? Fourth, the vehicle classification is not consistently used in the manuscript. For example, there are a number of vehicle types in Table S4 and S5 that are not described in Sect. 2.1. "Mini" truck is mentioned line 215, but is not described in Sect. 2.1. The authors classified the vehicles into passenger vehicles (LD, MD, and HD), taxis, buses, trucks (LD, MD, and HD), and motorcycles in Sect. 2.1. However, later in the main text (e.g,. Sec. 3.2, Figures 1a, 4c, and 5a, and Table S4 and S5), it seems that they also considered taxis and buses as passenger vehicles. If so, this should be stated in the manuscript, and classification criterions should be clearly provided.

Line 143, please double check whether provincial motorcycle population data are provided in China Automotive Industry Yearbook.

Sect. 2.2 and 3.1.2. Except for LDPVs, the authors did not provide any VKT data for all the other vehicle types. The review suggests the authors adding a table to summarize the VKT values of all vehicle types (i.e., LDPV, MDPV, HDPV, Taxi, Bus, LDT, MDT, HDT, and motorcycles) used in this work. The reviewer suggests the authors providing the vehicle population, not the population percentage, by vehicle type (i.e., LDPV, MDPV, HDPV, Taxi, Bus, LDT, MDT, HDT, and motorcycles) and by control technology (i.e., China 0 to 5) in Table 1. The population of motorcycle is missing in Table 1. In addition to Figures 4 and 5, the reviewer suggests the authors providing a table to summarize the VOC/IVOC emissions at the country level by vehicle type (i.e., LDPV, MDPV, HDPV, Taxi, Bus, LDT, MDT, HDT, and motorcycles) and by control technology (i.e., China 0 to 5).

About emission factors (EFs) of VOC/IVOC, the reviewer has the following suggestions and questions:

(1) Title of Table S4 and S5, indicate these are "tailpipe" VOCs. Please also double check whether the unit is mg/km or g/km

(2) Tailpipe EFs of motorcycles are missing in Table S4 and S5. They are not mentioned in the main text, either.

(3) The vehicle classification in Table S4 and Table S5 is different from the description in Sect. 2.1. For example, LDGTAs, LDDTAs, LDABs, MDGBUs, MDDBUs, MDABs, HDGBUs, HDDBUs, and HDABs, these vehicle types are not mentioned in Sect. 2.1, nor in the results and discussion section. If the study was conducted with more detailed vehicle classification, it should be introduced in the main text.

(4) The EFs of evaporation are not given. The reviewer suggests adding a table listing EFs of diurnal loss (<24, 24-48, and 48-72), hot soak, refueling, and running loss by vehicle type (i.e., LDPV, MDPV, HDPV, Taxi, Bus, LDT, MDT, HDT, and motorcycles). Data sources should be provided too.

[Figure]

(5) Line 239-248. First, the meanings of T, N, and P in Eqs. (3)-(6) are not provided. Second, besides simply providing the meanings of each variable in Eqs. (3)-(6), the authors are suggested to explain these equations.

(6) Line 244, 264, why China 4 LDGVs' EFs could be used for all non-motorcycle vehicle types and control technologies?

(7) Eqs. (7), (9), (11), (12). The authors claimed that the units of EFs are g/hour. The reviewer believes that this is not correct.

Line 290. Is the motor gasoline consumption by province calculated or derived from official statistics? Methods or data sources should be provided.

Main text after Sect. 3.2 may need to be polished to make it read like a scientific article.

The authors are suggested to check citations carefully before submitting the revised manuscript. Examples are:

Line 61, change "Cai et al" to "Cai and Xie". Remove "(Cai et al., 2009)" in line 62

Yang et al., 2015 is mentioned several times in the manuscript (e.g., lines 95, 106, 150, 194, 205, 281, etc.). However, there are two references by Yang et al. in 2015. Letters a and b should be added to the year both in the in-text citation as well as in the reference list.

Line 163-164, 179-180, "Zhao et al." to "Zhao et al. (2015, 2016)" and remove "(Zhao et al.; 2016; Zhao et al; 2015)"

Line 275, 307, "ICCT, 2012" is not in the reference list

Line 301, "MOVES, 2010" is not in the reference list

Line 326, "Man et al., 2016" is not in the reference list

In the reference list, there are lots of references that are not cited in the main text. Please have them carefully checked before submitting the revised manuscript.

Minor editorial issues:

Line 121, remove "five". According the introduction section, it seems that there are six deficiencies, while in Sect. 4, it seems the authors discussed four aspects.

Line 217, "POA" should be defined in the first appearance.

Line 257, "GTs"??

Line 324, incomplete sentence

Line 394, "eg." to "e.g., "

What is the unit of EFs in Table S3?

The caption of Figure 1 should be self-explained.

There are grammatical errors throughout the manuscript. I strongly suggest a grammar checking by native English speaker before submitting the revised manuscript. Examples in the first five pages are: Abstract should be written in the present tense.

Line 41-42.

Line 47, remove "the year of"

Line 62, add "during" after "China"

Line 63, "include" to "included", add "a" before "part"

Line 68, "provide" o "provided"

Line 70 remove "trend"

Line 74, "has" to "have", "a non-ignorable contributor" to "non-ignorable contributors"

Line 76

Line 81, "profile" to "profiles"

Line 82, "with" to "to"

Line 83, "were" to "are", "method section" to "Sect. 2"

Line 84, "impact" to "impacts", "atmospheric condition" to "air quality"???

Line 86, "complicate" to "complicated", add "of" after "a series"

Line 90, "measurements" to "measurement", "none of the" to "to our knowledge, there is no", add "for China" at the end of this sentence

Line 98

Line 100, "method" to "methods"

Line 106, "emission" to "emissions were"

Line 109, "common-used" to "commonly-used"

Line 111, "provided" to "provide", "level" to "levels"

Line 113, "recently" to "recent".

Line 114, add "furthermore," at the beginning of the sentence, "provides" to "provide", "types" to "type"

Line 115, remove "However,"

Line 116-117, change to "More detailed vehicle population data by fuel type and by control technology are required to calculate emissions because they have been reported to ..."

Line 120, "were" to "are"

Line 121, "were" to "are"

Line 123, "were" to "are"

Line 124, change to "there is no local IVOC emission factor reported"

There are more...

---

## Referee Comment (RC2) · Anonymous Referee #2 · 16 Aug 2017

General Comments:

This work developed an updated speciated emission inventory of VOCs and IVOCs from vehicles in China for the year of 2015 based on a set of state-of-the-art methodologies and a mass of local measurement data. The strength of this inventory is that massive GPS records and questionnaire analysis are collected to better characterize the activity level. In addition, in terms of the method, this work improved the emission estimation by including evaporative emission calculation and applying road emission intensity based approach. This well-written and well-structured paper is potentially important and will be valuable in the future for modelling the formation of fine particles and ozone pollution in China. There are a few comments that need to be addressed to improve the paper and make it more accessible to the wide audience of users of the

information presented.

Specific Comments:

In the first place, the information need to be made available, for example through the journal with a doi, or through the website of the author's institute.

A second recommendation is that speciated emission inventory of VOCs and IVOCs based on prevailing lumped chemical mechanisms like CB05 and SAPRC are suggested to be provided since that this emission database will be mainly used in chemical transport models.

There still large uncertainty lies in activity level, emission factor and the estimation method itself. Another recommendation is that uncertainty analysis ought to be conducted and more quantitative results should be provided in Section 3.3.

Technical Corrections:

Section 3.2.2-3.2.4 are too short to be an individual section. I personally think that this part of discussion is not necessarily to be divided into three sections.

Supporting Information, Table S4: Some abbreviations of vehicle types (LDGTAs, LD-DTAs) ought to be specified.

Some in-text citations are missing in the reference list, e.g., MOVES, 2010; ICCT, 2012.

---

## Author Comment (AC1) · 30 Aug 2017

In this work, Liu et al. developed an updated VOC emission inventory for the on road vehicles in China for 2015. Particularly, they refined their analysis by using vehicle activity data collected from a large number of GPS records, taking into account VOC evaporation emissions from gasoline vehicles, and including tailpipe IVOC emission estimates. The topic is suitable for the Atmospheric Chemistry and Physics and the technical part of the manuscript is relatively well described. However, there are some technical, editorial, and grammatical issues in the current manuscript that need to be clarified and corrected first. The manuscript is highly suggested to be grammar checked by native English speakers. Considering the results of this work is potentially of great value to the atmospheric modeling community, the reviewer suggests a carefully revised manuscript for publication in ACP.

**Response:** We have revised as your comments point-by-point. The manuscript was carefully reviewed by two native speakers. We carefully response all technical comments and the provide as much as details and raw data.

Thank you for the help!

Abstract line 27. The authors state in the abstract that the VKT level of "trucks were calculated from reported data by more than 2 million trucks in China". According to the Chinese official statistics, there were ~20 million trucks in China in 2015. That means the authors have collected VKT data of ~10%, which is a quit decent sampling ratio, of all trucks in China. However, relevant results are neither described nor referenced to previous studies in the main text (e.g., Sect. 2.2 and 3.1.2). The reviewer suggests the authors adding the description and discussion paragraphs or sections to introduce the methods and results in detail.

**Response:** We have added discussion and description on trucks' VKT data in Section 2.2. Those data were purchased from commercial big data platform combining with our survey to get the spatial distribution. The platform is the official service provider for all commercial trucks in China under Ministry of Transportation. Due to the licenses in contract, we could not provide raw data to the third party at this stage. Thus, the details of VKT were not released.

**Author's changes in manuscript:**

In section 2.2 Vehicle activity:

"The average mileage for trucks were obtained from a commercial source with data feeding of more than 2 million trucks, mainly commercial vehicles installed with either the GPS or China Bei Dou System (BDS). Location, speed and vehicle type information are live fed to the commercial platform. The VKT for each truck category was calculated using the monitored data from the platform."

In section 3.1.2 VKT characteristics of LDPVs:

Table 3 summarized vehicle mileages of trucks in China. VKT of trucks is significantly influenced by vehicle age. The annual mileage of China 0 and China 1 trucks are much lower than vehicles of the same type with better control technologies. Aging of trucks greatly impact their performances due to common overloading seen in China. Several cities have implemented low emission zones to restrict entry of trucks with outdated control technologies.

**Table 3 Average annual VKT in China (Km/year)**

|         | LDGTs | LDDTs | MDGTs | MDDTs | HDGTs | HDDTs | TAs    | BUs   | MDPVs | LDPVs  |
|---------|-------|-------|-------|-------|-------|-------|--------|-------|-------|--------|
| China 0 | 22160 | 19270 | 35196 | 21231 | 27716 | 24372 | 138000 | 50000 | 31300 | 114800 |
| China 1 | 22160 | 19270 | 35196 | 21231 | 27716 | 24372 |        |       |       |        |
| China 2 | 26335 | 26964 | 40766 | 28140 | 33226 | 38485 |        |       |       |        |
| China 3 | 29467 | 36581 | 47927 | 36366 | 40310 | 64128 |        |       |       |        |
| China 4 | 34165 | 45237 | 53497 | 60308 | 45820 | 98206 |        |       |       |        |
| China 5 | 34165 | 45237 | 53497 | 60308 | 45820 | 98206 |        |       |       |        |

The section heading of Sect. 2.1 is "Vehicle stock and classification". However, only vehicle classification is described.

**Response:** The method of getting vehicle stock was discussed in the revised section 2.1. The vehicle stock data was discussed in Section 3.1.

**Author's changes in manuscript:** The following sentences were added.

Detailed provincial population data of all vehicles excluding GMs in 2015 was obtained by complete statistical survey conducted by the Vehicle Emission Control Center (VECC) of China's Ministry of Environmental Protection (MEP), which could be considered highly accurate. The provincial GMs population in 2015 was obtained from the Provincial Statistic Yearbook 2016 of each province.

The classification of vehicles is not very clear to the reviewer and needs more clarification. First, the criterions to distinguish LD, MD, and HD passenger vehicles and to distinguish LD, MD, and HD trucks are not given. Second, if taxis are classified

as separate vehicle types, the authors should add a statement previously that LD, MD, and HD passenger vehicles do not include taxis. Third, the reviewer is wondering what kind of vehicles are treated as alternative-fuel vehicles? Electric? Plug-in electric? Hybrid? Internal combustion engine vehicles running on alternative fuels such as CNG/LNG/LPG, methanol, or ethanol? Fourth, the vehicle classification is not consistently used in the manuscript. For example, there are a number of vehicle types in Table S4 and S5 that are not described in Sect. 2.1. "Mini" truck is mentioned line 215, but is not described in Sect. 2.1. The authors classified the vehicles into passenger vehicles (LD, MD, and HD), taxis, buses, trucks (LD, MD, and HD), and motorcycles in Sect. 2.1. However, later in the main text (e.g., Sec. 3.2, Figures 1a, 4c, and 5a, and Table S4 and S5), it seems that they also considered taxis and buses as passenger vehicles. If so, this should be stated in the manuscript, and classification criterions should be clearly provided.

**Response:** The whole section was rewritten to be clear. The revised section has addressed all concerns from reviewer. The new classification is keeping consistent through the whole manuscript as well as the supporting information. The criterions to distinguish LD, MD, and HD passenger vehicles and to distinguish LD, MD, and HD trucks are given. Second, we provided a statement that LD, MD, and HD passenger vehicles do not include taxis. Third, we discussed what kind of vehicles are treated as alternative-fuel vehicles.

**Author's changes in manuscript:** The following sentences were added.

"In total, 25 types of on-road vehicles were considered in this study, including passenger vehicles, trucks and motorcycles (GMs). Passenger vehicles were further divided into 18 types: light-duty gasoline passenger vehicles excluding taxies (LDGPVs), light-duty diesel passenger vehicles excluding taxies (LDDPVs), light-duty alternative-fuel passenger vehicles excluding taxies (LDAPVs), medium-duty gasoline passenger vehicles excluding buses (MDGPVs), medium-duty diesel passenger vehicles excluding buses (MDDPVs), medium-duty alternative-fuel passenger vehicles excluding buses (MDAPVs), heavy-duty gasoline passenger vehicles excluding buses (HDGPVs), heavy-duty diesel passenger vehicles excluding buses (HDDPVs), heavy-duty alternative-fuel passenger vehicles excluding buses (HDAPVs), light-duty gasoline taxis (LDGTAs), light-duty diesel taxis (LDDTAs), light-duty alternative-fuel taxis (LDATAs), medium-duty gasoline buses (MDGBUs), medium-duty diesel buses (MDDBUs), medium-duty alternative-fuel buses (MDABUs), heavy-duty gasoline buses (HDGBUs), heavy-duty diesel buses (HDDBUs) and heavy-duty alternative-fuel buses (HDABUs). For passenger vehicles, light-duty refers to vehicles with length less than 6000mm and ridership no more than 9. Medium-duty refers to vehicles of length less than 6000mm and ridership between 10-19. Heavy-duty refers to vehicles of length no less than 6000mm or ridership is no less than 20. These vehicles were further classified by control technologies (i.e., China 0, China 1, China 2, China 3, China 4 and above). Alternative-fuel vehicles in this study include compressed natural gas (CNG),

liquefied natural gas (LNG) and liquefied petroleum gas (LPG) vehicles.

Trucks (or freight trucks) were divided into 6 types: light-duty gasoline trucks (LDGTs), light-duty diesel trucks (LDDTs), medium-duty gasoline trucks (MDGTs), medium-duty diesel trucks (MDDTs), heavy-duty gasoline trucks (HDGTs), heavy-duty diesel trucks (HDDTs). For trucks, a light-duty truck refers vehicles with mass less than 3500kg. A medium-duty truck refers to vehicles with mass ranging from 3500kg to 12000kg. A heavy-duty truck refers vehicles of mass more than 12000kg."

Line 143, please double check whether provincial motorcycle population data are provided in China Automotive Industry Yearbook.

**Response:** The provincial GMs population in 2015 was obtained from the Provincial Statistic Yearbook 2016 of each province.

**Author's changes in manuscript:** The provincial GMs population in 2015 was obtained from the Provincial Statistic Yearbook 2016 of each province.

Sect. 2.2 and 3.1.2. Except for LDPVs, the authors did not provide any VKT data for all the other vehicle types. The review suggests the authors adding a table to summarize the VKT values of all vehicle types (i.e., LDPV, MDPV, HDPV, Taxi, Bus, LDT, MDT, HDT, and motorcycles) used in this work. The reviewer suggests the authors providing the vehicle population, not the population percentage, by vehicle type (i.e., LDPV, MDPV, HDPV, Taxi, Bus, LDT, MDT, HDT, and motorcycles) and by control technology (i.e., China 0 to 5) in Table 1. The population of motorcycle is missing in Table 1. In addition to Figures 4 and 5, the reviewer suggests the authors providing a table to summarize the VOC/IVOC emissions at the country level by vehicle type (i.e., LDPV, MDPV, HDPV, Taxi, Bus, LDT, MDT, HDT, and motorcycles) and by control technology (i.e., China 0 to 5).

**Response:**
(1) Beside the VKT data for LDPV, which was discussed in detail, the VKT for all the other vehicle types were summarized in Table 3.
(2) Table 1 was revised and now provides vehicle population instead of percentage.
(3) The population of motorcycle was added into Table 1 and the main text in Sect. 3.1.1.
(4) We also added two tables including detailed tailpipe VOC/IVOC emissions by vehicle type and by control technology. For evaporative emissions, it's not calculated based on the vehicle type or control technology. We could not distribute the total gasoline consumption into these categories. Thus, no such data was provided further than Figure 4.

**Author's changes in manuscript:**

(1) For VKT:

Table 3 summarized vehicle mileages of other vehicle types in China. VKT of trucks is significantly influenced by vehicle age. The annual mileage of China 0 and China 1 trucks are much lower than vehicles of the same type with better control technologies. Aging of trucks greatly impact their performances due to common overloading seen in China. Several cities have implemented low emission zones to restrict entry of trucks with outdated control technologies.

**Table 3 Average annual VKT in China (Km/year)**

|  | LDGTs | LDDTs | MDGTs | MDDTs | HDGTs | HDDTs | TAs | BUs | MDPVs | LDPVs |
|---|---|---|---|---|---|---|---|---|---|---|
| **China 0** | 22160 | 19270 | 35196 | 21231 | 27716 | 24372 | 138000 | 50000 | 31300 | 114800 |
| **China 1** | 22160 | 19270 | 35196 | 21231 | 27716 | 24372 | | | | |
| **China 2** | 26335 | 26964 | 40766 | 28140 | 33226 | 38485 | | | | |
| **China 3** | 29467 | 36581 | 47927 | 36366 | 40310 | 64128 | | | | |
| **China 4** | 34165 | 45237 | 53497 | 60308 | 45820 | 98206 | | | | |
| **China 5** | 34165 | 45237 | 53497 | 60308 | 45820 | 98206 | | | | |

(2) For population:

Table 1 summarized the vehicle population and corresponding proportions classified by fuel types. LDPVs, MDPVs and TAs were mainly fuelled by gasoline while HDPVs, LDTs, MDTs, HDTs and BUs were primarily fuelled by diesel.

**Table 1 Population of different types of vehicles in China in 2015**

| Vehicle type | Population | Fuel type percentage (%) | | | Control technology | Population |
|---|---|---|---|---|---|---|
| | | Gasoline | Diesel | Alternative fuels | | |
| LDPVs | 137599368 | 97.96 | 1.15 | 0.90 | China 0 | 7062516 |
| MDPVs | 1428102 | 56.53 | 40.68 | 2.78 | China 1 | 16181788 |
| HDPVs | 1165836 | 15.97 | 75.03 | 9.00 | China 2 | 12251006 |
| LDTs | 15998479 | 41.50 | 58.50 | 0.00 | China 3 | 86584457 |
| MDTs | 2826881 | 18.92 | 81.08 | 0.00 | China 4 | 38880534 |
| HDTs | 6037719 | 7.65 | 92.35 | 0.00 | China 5 | 8834416 |
| TAs | 3910397 | 61.89 | 29.37 | 8.74 | | |
| BUs | 827935 | 13.76 | 55.39 | 30.85 | | |
| GMs | 88759010 | 100 | 0 | 0 | | |

(3) For motorcycle population:

GMs and non-GM vehicles contributed 34.3% (88,759,010) and 65.7% (169,794,718) respectively (Figure 1) among the 259 million total on-road vehicles in China in the

year 2015.

(4) For emissions:

**Table 4 VOC tailpipe emissions by vehicle type and by control technology in China in 2015 (Gg)**

|  | China 0 | China 1 | China 2 | China 3 | China 4 | China 5 | SUM |
|---|---|---|---|---|---|---|---|
| LDPVs | 173.59 | 146.09 | 56.48 | 240.32 | 49.09 | 8.81 | 674.38 |
| MDPVs | 56.73 | 10.28 | 7.42 | 4.88 | 0.62 | 0.06 | 79.98 |
| HDPVs | 99.57 | 22.13 | 24.31 | 45.37 | 5.72 | 2.12 | 199.23 |
| LDTs | 25.15 | 41.60 | 13.52 | 85.70 | 6.52 | 0.03 | 172.52 |
| MDTs | 42.20 | 12.18 | 1.16 | 7.90 | 0.50 | 0.01 | 63.95 |
| HDTs | 44.34 | 10.83 | 2.47 | 56.38 | 5.42 | 0.21 | 119.65 |
| TAs | 97.44 | 71.43 | 50.55 | 74.33 | 15.30 | 2.06 | 311.12 |
| BUs | 5.25 | 1.65 | 3.43 | 1.52 | 0.09 | 0.05 | 11.99 |
| GMs |  |  |  |  |  |  | 563.18 |

**Table 5 IVOC tailpipe emissions by vehicle type and by control technology in China in 2015 (Gg)**

|  | China 0 | China 1 | China 2 | China 3 | China 4 | China 5 | SUM |
|---|---|---|---|---|---|---|---|
| LDPVs | 5.07 | 18.05 | 1.72 | 10.11 | 2.59 | 0.23 | 37.76 |
| MDPVs | 0.31 | 0.09 | 0.02 | 0.06 | 0.01 | 0.00 | 0.51 |
| HDPVs | 0.40 | 0.28 | 0.07 | 0.33 | 0.26 | 0.00 | 1.33 |
| LDTs | 1.58 | 2.66 | 0.61 | 18.38 | 2.82 | 0.02 | 26.07 |
| MDTs | 2.01 | 0.77 | 0.27 | 4.86 | 0.60 | 0.01 | 8.51 |
| HDTs | 1.48 | 1.19 | 0.44 | 27.68 | 5.25 | 0.21 | 36.26 |
| TAs | 1.97 | 5.77 | 0.49 | 2.27 | 0.22 | 0.01 | 10.73 |
| BUs | 0.02 | 0.02 | 0.01 | 0.02 | 0.00 | 0.00 | 0.07 |

About emission factors (EFs) of VOC/IVOC, the reviewer has the following suggestions and questions:

(1) Title of Table S4 and S5, indicate these are "tailpipe" VOCs. Please also double check whether the unit is mg/km or g/km

**Response:** The titles of Table S4 and S5 were revised and the units are all g/km.

**Author's changes in manuscript:**
Table S4. VOCs tailpipe emission factors used in this study (g/km).
Table S5. IVOCs tailpipe emission factors used in this study (g/km).

(2) Tailpipe EFs of motorcycles are missing in Table S4 and S5. They are not mentioned in the main text, either.

**Response:** Tailpipe EFs of motorcycles are added in Table S4. The IVOC emissions were only calculated for non-GMs. We revised the sentences in Sect. 2.3 to clarify this point.

**Author's changes in manuscript:**

For IVOCs emission factors, a mapping to match US emission certification level to China emission level was built (Table S3). Only the non-GMs were considered for the IVOC emissions evaluation.

**Table S4. VOCs tailpipe emission factors used in this study (g/km).**

| | Passenger vehicles | | | | | |
| | China 0 | China 1 | China 2 | China 3 | China 4 | China 5 |
|---|---|---|---|---|---|---|
| LDGTAs | 3.840 | 1.368 | 0.963 | 0.454 | 0.277 | 0.257 |
| LDDTAs | 0.785 | 0.071 | 0.046 | 0.024 | 0.016 | 0.016 |
| LDATAs | 3.788 | 0.433 | 0.398 | 0.115 | 0.066 | 0.293 |
| LDGPVs | 2.685 | 0.663 | 0.314 | 0.191 | 0.075 | 0.056 |
| LDDPVs | 0.785 | 0.071 | 0.046 | 0.024 | 0.016 | 0.016 |
| LDAPVs | 2.236 | 0.236 | 0.164 | 0.094 | 0.062 | 0.091 |
| MDGBUs | 5.144 | 5.255 | 1.980 | 0.869 | 0.418 | 0.418 |
| MDDBUs | 2.668 | 0.576 | 0.351 | 0.283 | 0.107 | 0.054 |
| MDABUs | 3.840 | 3.200 | 2.860 | 1.720 | 1.192 | 1.192 |
| MDGPVs | 3.695 | 2.567 | 1.443 | 0.373 | 0.107 | 0.107 |
| MDDPVs | 1.493 | 1.425 | 0.425 | 0.364 | 0.383 | 0.383 |
| MDAPVs | 1.920 | 1.600 | 1.430 | 0.860 | 0.596 | 0.596 |
| HDGBUs | 5.144 | 5.255 | 1.980 | 0.869 | 0.418 | 0.418 |
| HDDBUs | 2.668 | 0.576 | 0.351 | 0.283 | 0.107 | 0.054 |
| HDABUs | 3.840 | 3.200 | 2.860 | 1.720 | 1.192 | 1.192 |
| HDGPVs | 5.144 | 5.255 | 1.980 | 0.869 | 0.418 | 0.418 |
| HDDPVs | 2.668 | 0.576 | 0.351 | 0.283 | 0.107 | 0.054 |
| HDAPVs | 3.840 | 3.200 | 2.860 | 1.720 | 1.192 | 1.192 |

| | | Trucks | | | | |
| | | China 0 | China 1 | China 2 | China 3 | China 4/ 5 |
|---|---|---|---|---|---|---|
| Urban road | LDGTs | 5.391 | 3.593 | 2.389 | 0.637 | 0.176 |
| | LDDTs | 2.267 | 2.205 | 1.411 | 0.384 | 0.194 |
| | MDGTs | 7.441 | 7.326 | 3.268 | 1.482 | 0.619 |
| | MDDTs | 4.863 | 1.742 | 0.455 | 0.219 | 0.111 |
| | HDGTs | 7.295 | 7.306 | 3.249 | 1.464 | 0.600 |
| | HDDTs | 4.413 | 0.970 | 0.562 | 0.276 | 0.139 |
| Provincial road | LDGTs | 4.040 | 2.693 | 1.841 | 0.530 | 0.147 |
| | LDDTs | 1.699 | 1.653 | 1.087 | 0.320 | 0.162 |

| | | | | | |
|---|---|---|---|---|---|
| | MDGTs | 5.577 | 5.490 | 2.449 | 1.111 | 0.464 |
| | MDDTs | 3.645 | 1.306 | 0.341 | 0.164 | 0.083 |
| | HDGTs | 5.467 | 5.475 | 2.435 | 1.097 | 0.450 |
| | HDDTs | 3.308 | 0.727 | 0.421 | 0.207 | 0.105 |
| National road | LDGTs | 4.376 | 2.916 | 1.924 | 0.549 | 0.152 |
| | LDDTs | 1.840 | 1.790 | 1.136 | 0.331 | 0.167 |
| | MDGTs | 6.040 | 5.946 | 2.652 | 1.203 | 0.503 |
| | MDDTs | 3.947 | 1.414 | 0.369 | 0.178 | 0.090 |
| | HDGTs | 5.921 | 5.930 | 2.637 | 1.188 | 0.487 |
| | HDDTs | 3.582 | 0.787 | 0.456 | 0.224 | 0.113 |
| Freeway | LDGTs | 4.119 | 2.745 | 1.837 | 0.536 | 0.148 |
| | LDDTs | 1.732 | 1.685 | 1.085 | 0.323 | 0.163 |
| | MDGTs | 5.685 | 5.597 | 2.497 | 1.132 | 0.473 |
| | MDDTs | 3.716 | 1.331 | 0.348 | 0.168 | 0.085 |
| | HDGTs | 5.574 | 5.582 | 2.483 | 1.118 | 0.458 |
| | HDDTs | 3.372 | 0.741 | 0.429 | 0.211 | 0.107 |
| County road | LDGTs | 7.010 | 4.673 | 3.059 | 0.798 | 0.221 |
| | LDDTs | 2.948 | 2.868 | 1.806 | 0.482 | 0.243 |
| | MDGTs | 9.677 | 9.527 | 4.250 | 1.927 | 0.805 |
| | MDDTs | 6.324 | 2.266 | 0.592 | 0.285 | 0.145 |
| | HDGTs | 9.487 | 9.501 | 4.226 | 1.903 | 0.780 |
| | HDDTs | 5.740 | 1.261 | 0.731 | 0.358 | 0.181 |
| **GMs** | | | | | | |
| GMs | | | 1.269 | | | |

(3) The vehicle classification in Table S4 and Table S5 is different from the description in Sect. 2.1. For example, LDGTAs, LDDTAs, LDABs, MDGBUs, MDDBUs, MDABs, HDGBUs, HDDBUs, and HDABs, these vehicle types are not mentioned in Sect. 2.1, nor in the results and discussion section. If the study was conducted with more detailed vehicle classification, it should be introduced in the main text.

**Response:** Yes. The calculation is based on more detailed classification. We have modified Sect. 2.1. Now the vehicle classification is consistent through the whole manuscript.

**Author's changes in manuscript:**

In total, 25 types of on-road vehicles were considered in this study, including passenger vehicles, trucks and motorcycles (GMs). Passenger vehicles were further divided into 18 types: light-duty gasoline passenger vehicles excluding taxies (LDGPVs), light-duty diesel passenger vehicles excluding taxies (LDDPVs), light-duty alternative-fuel passenger vehicles excluding taxies (LDAPVs), medium-duty gasoline passenger

vehicles excluding buses (MDGPVs), medium-duty diesel passenger vehicles excluding buses (MDDPVs), medium-duty alternative-fuel passenger vehicles excluding buses (MDAPVs), heavy-duty gasoline passenger vehicles excluding buses (HDGPVs), heavy-duty diesel passenger vehicles excluding buses (HDDPVs), heavy-duty alternative-fuel passenger vehicles excluding buses (HDAPVs), light-duty gasoline taxis (LDGTAs), light-duty diesel taxis (LDDTAs), light-duty alternative-fuel taxis (LDATAs), medium-duty gasoline buses (MDGBUs), medium-duty diesel buses (MDDBUs), medium-duty alternative-fuel buses (MDABU), heavy-duty gasoline buses (HDGBUs), heavy-duty diesel buses (HDDBUs) and heavy-duty alternative-fuel buses (HDABUs). For passenger vehicles, light-duty refers to vehicles with length less than 6000mm and ridership no more than 9. Medium-duty refers to vehicles of length less than 6000mm and ridership between 10-19. Heavy-duty refers to vehicles of length no less than 6000mm or ridership is no less than 20. These vehicles were further classified by control technologies (i.e., China 0, China 1, China 2, China 3, China 4 and above). Alternative-fuel vehicles in this study include compressed natural gas (CNG), liquefied natural gas (LNG) and liquefied petroleum gas (LPG) vehicles.

Trucks (or freight trucks) were divided into 6 types: light-duty gasoline trucks (LDGTs), light-duty diesel trucks (LDDTs), medium-duty gasoline trucks (MDGTs), medium-duty diesel trucks (MDDTs), heavy-duty gasoline trucks (HDGTs), heavy-duty diesel trucks (HDDTs). For trucks, a light-duty truck refers vehicles with mass less than 3500kg. A medium-duty truck refers to vehicles with mass ranging from 3500kg to 12000kg. A heavy-duty truck refers vehicles of mass more than 12000kg.

| Passenger vehicles | | | | | | |
|---|---|---|---|---|---|---|
| | China 0 | China 1 | China 2 | China 3 | China 4 | China 5 |
| LDGTAs | 3.840 | 1.368 | 0.963 | 0.454 | 0.277 | 0.257 |
| LDDTAs | 0.785 | 0.071 | 0.046 | 0.024 | 0.016 | 0.016 |
| LDATAs | 3.788 | 0.433 | 0.398 | 0.115 | 0.066 | 0.293 |
| LDGPVs | 2.685 | 0.663 | 0.314 | 0.191 | 0.075 | 0.056 |
| LDDPVs | 0.785 | 0.071 | 0.046 | 0.024 | 0.016 | 0.016 |
| LDAPVs | 2.236 | 0.236 | 0.164 | 0.094 | 0.062 | 0.091 |
| MDGBUs | 5.144 | 5.255 | 1.980 | 0.869 | 0.418 | 0.418 |
| MDDBUs | 2.668 | 0.576 | 0.351 | 0.283 | 0.107 | 0.054 |
| MDABUs | 3.840 | 3.200 | 2.860 | 1.720 | 1.192 | 1.192 |
| MDGPVs | 3.695 | 2.567 | 1.443 | 0.373 | 0.107 | 0.107 |
| MDDPVs | 1.493 | 1.425 | 0.425 | 0.364 | 0.383 | 0.383 |
| MDAPVs | 1.920 | 1.600 | 1.430 | 0.860 | 0.596 | 0.596 |
| HDGBUs | 5.144 | 5.255 | 1.980 | 0.869 | 0.418 | 0.418 |
| HDDBUs | 2.668 | 0.576 | 0.351 | 0.283 | 0.107 | 0.054 |
| HDABUs | 3.840 | 3.200 | 2.860 | 1.720 | 1.192 | 1.192 |
| HDGPVs | 5.144 | 5.255 | 1.980 | 0.869 | 0.418 | 0.418 |
| HDDPVs | 2.668 | 0.576 | 0.351 | 0.283 | 0.107 | 0.054 |
| HDAPVs | 3.840 | 3.200 | 2.860 | 1.720 | 1.192 | 1.192 |
| Trucks | | | | | | |
| | China 0 | China 1 | China 2 | China 3 | China 4/ 5 | |

| | | | | | |
|---|---|---|---|---|---|
| | LDGTs | 5.391 | 3.593 | 2.389 | 0.637 | 0.176 |
| | LDDTs | 2.267 | 2.205 | 1.411 | 0.384 | 0.194 |
| Urban road | MDGTs | 7.441 | 7.326 | 3.268 | 1.482 | 0.619 |
| | MDDTs | 4.863 | 1.742 | 0.455 | 0.219 | 0.111 |
| | HDGTs | 7.295 | 7.306 | 3.249 | 1.464 | 0.600 |
| | HDDTs | 4.413 | 0.970 | 0.562 | 0.276 | 0.139 |
| | LDGTs | 4.040 | 2.693 | 1.841 | 0.530 | 0.147 |
| | LDDTs | 1.699 | 1.653 | 1.087 | 0.320 | 0.162 |
| Provincial road | MDGTs | 5.577 | 5.490 | 2.449 | 1.111 | 0.464 |
| | MDDTs | 3.645 | 1.306 | 0.341 | 0.164 | 0.083 |
| | HDGTs | 5.467 | 5.475 | 2.435 | 1.097 | 0.450 |
| | HDDTs | 3.308 | 0.727 | 0.421 | 0.207 | 0.105 |
| | LDGTs | 4.376 | 2.916 | 1.924 | 0.549 | 0.152 |
| | LDDTs | 1.840 | 1.790 | 1.136 | 0.331 | 0.167 |
| National road | MDGTs | 6.040 | 5.946 | 2.652 | 1.203 | 0.503 |
| | MDDTs | 3.947 | 1.414 | 0.369 | 0.178 | 0.090 |
| | HDGTs | 5.921 | 5.930 | 2.637 | 1.188 | 0.487 |
| | HDDTs | 3.582 | 0.787 | 0.456 | 0.224 | 0.113 |
| | LDGTs | 4.119 | 2.745 | 1.837 | 0.536 | 0.148 |
| | LDDTs | 1.732 | 1.685 | 1.085 | 0.323 | 0.163 |
| Freeway | MDGTs | 5.685 | 5.597 | 2.497 | 1.132 | 0.473 |
| | MDDTs | 3.716 | 1.331 | 0.348 | 0.168 | 0.085 |
| | HDGTs | 5.574 | 5.582 | 2.483 | 1.118 | 0.458 |
| | HDDTs | 3.372 | 0.741 | 0.429 | 0.211 | 0.107 |
| | LDGTs | 7.010 | 4.673 | 3.059 | 0.798 | 0.221 |
| | LDDTs | 2.948 | 2.868 | 1.806 | 0.482 | 0.243 |
| County road | MDGTs | 9.677 | 9.527 | 4.250 | 1.927 | 0.805 |
| | MDDTs | 6.324 | 2.266 | 0.592 | 0.285 | 0.145 |
| | HDGTs | 9.487 | 9.501 | 4.226 | 1.903 | 0.780 |
| | HDDTs | 5.740 | 1.261 | 0.731 | 0.358 | 0.181 |
| **GMs** | | | | | | |
| GMs | | | | 1.269 | | |

**Table S5. IVOCs tailpipe emission factors used in this study (g/km).**

| Passenger vehicles | | | | | |
|---|---|---|---|---|---|
| | China 0/1 | China 2 | China 3 | China 4 | China 5 |
| LDDTAs/LDGTAs/LDGPVs/LDDPVs | **0.09287[1]** | **0.00977** | **0.00809** | **0.00413** | **0.00151** |
| MDGBUs/MDDBUs/MDGPVs/MDDPVs | **0.01837** | **0.00424** | **0.00532** | **0.00532** | **0.00221** |
| HDGBUs/HDDBUs/HDGPVs/HDDPVs | **0.01671** | **0.00447** | **0.00447** | **0.02553** | **0.00231** |
| **Trucks** | | | | | |
| | China 0/1 | China 2 | China 3 | China 4 | China 5 |
| LDGTs | **0.07200** | **0.00266** | **0.00266** | **0.00333** | **0.00272** |

| | | | | | |
|---|---|---|---|---|---|
| LDDTs | 0.06072[2] | 0.06072 | 0.06072 | 0.08574 | 0.08574 |
| MDGTs | 0.10800 | 0.00399 | 0.00399 | 0.00500 | 0.00409 |
| MDDTs | 0.09108 | 0.09108 | **0.09108** | **0.12861** | 0.01122 |
| HDGTs | 0.10800 | 0.00399 | 0.00399 | 0.00500 | 0.00409 |
| HDDTs | 0.34478 | 0.34478 | 0.34478 | **0.34478** | **0.01122** |

[1] The bold fonts mean that data is from measurements in literature. It is equal to the median of measurements for all samples in this vehicle category.

[2] The non-bold fonts mean that no measurement data is available. The emission factor is derived based on the following assumptions: EF(HD)=EF(MD)=1.5*EF(LD) and EF (control level) = EF (control level±n, where measurement data is available).

(4) The EFs of evaporation are not given. The reviewer suggests adding a table listing EFs of diurnal loss, hot soak, refueling, and running loss by vehicle type (i.e., LDPV, MDPV, HDPV, Taxi, Bus, LDT, MDT, HDT, and motorcycles). Data sources should be provided too

**Response:** A table including EFs of evaporation is added in supporting information. The data sources are also provided.

**Author's changes in manuscript:**
The emission factors of diurnal and hot soak were obtained by a set of Sealed Housing for Evaporative Determination (SHED) tests, as was introduced in our previous study (*Liu et al., 2015*). The detailed emission factors were summarized in Table S6.

For motorcycles, the calculation of evaporative emissions was simplified. Because the activity data could not support to calculate diurnal, refueling, hot soak or running loss. So we use the following equation to calculate total evaporative emissions for GMs based on the mileages.

$$E_{GMs,i} = EF_{GMs} \times VP_{i,GMs} \times VKT_{i,GMs} \ , \qquad\qquad (7)$$

where $E_{GMs,i}$ represents the annual evaporative emissions from GMs registered in province $i$ (g·year$^{-1}$); $EF_{GMs}$ represents the evaporative emission factor of GMs (g·km$^{-1}$); For VEEs from GMs, the emission factors given by the International Council on Clean Transportation (ICCT) were utilized (*ICCT, 2012*). $VKT_{i,GMs}$ represents the annual VKT of GMs in province $i$ (km·year$^{-1}$).

**Table S6. Evaporation emission factors used in this study.**

| | | Parking duration | Unite | Emission factors |
|---|---|---|---|---|
| vehicles | Diurnal | <24 hour | g/hour | 0.094[1] |
| | | 24-48 hour | g/hour | 0.247[1] |

| | | | |
|---|---|---|---|
| | >48 hour | g/hour | 0.339[1] |
| | Hot soak | g/hour | 0.083[1] |
| | Refueling (without control) | g/L | 0.848[1] |
| | Running loss | g/hour | 11.6[2] |
| motorcycle | | g/km | 0.57[3] |

References:

1. Liu, H.; Man, H.; Tschantz, M.; Wu, Y.; He, K.; Hao, J., VOC from Vehicular Evaporation Emissions: Status and Control Strategy. *Environ. Sci. Technol* **2015**, 49, (24), 14424-14431. DOi:10.1021/acs.est.5b04064

2. EPA-420-R-12-027; Development of Evaporative Emissions Calculations for the Motor Vehicle Emissions Simulator MOVES2010; United States Environmental Protection Agency; Washington, DC, **2012**; https://nepis.epa.gov/Exe/ZyPDF.cgi/P100F3ZY.PDF?Dockey=P100F3ZY.PDF

3. ICCT, Air Emissions Issues Related to Two and Three-Wheeled Motor Vehicles an Initial Assessment of Current Conditions and Options for Control; International Council on Clean Transportation (ICCT), 2007; http://www.theicct.org/sites/default/files/publications/twothree_wheelers_2007.pdf

(5) Line 239-248. First, the meanings of T, N, and P in Eqs. (3)-(6) are not provided. Second, besides simply providing the meanings of each variable in Eqs. (3)-(6), the authors are suggested to explain these equations.

**Response:** The meanings of each parameter were provided now. Some sentences were added above the equations to explain the calculation.

**Author's changes in manuscript:**

"For diurnal emissions, we calculated total parking hours for each parking events and adjust emissions based on how long the vehicle was parked. The first hour for each parking event was treated as the hot soak and was subtracted from the diurnal emissions."

"For diurnal emissions, we calculated total parking hours for each parking events and adjust emissions based on how long the vehicle was parked. The first hour for each parking event was treated as the hot soak and was subtracted from the diurnal emissions."

"According to the US EPA, hot soak is defined as the evaporative losses that occur within the one-hour period after the engine is shut down (EPA420-R-01-026). If the parking duration is longer than one hour, then the extra vapor losses fall into diurnal emissions. The provincial hot soak emissions for non-GM gasoline vehicles (i.e., LDGPVs, MDGPVs, HDGPVs, LDGTAs , GBUs, LDGTs, MDGTs, HDGTs) were

calculated by Eq. (8):"

"China is following European control experiences to popularize Stage-II vapor control system in refuelling stations to reduce refuelling loss. The vehicle refuelling emissions were also measured by our team from SHED tests (Yang et al, 2015b). The provincial refuelling emissions from gasoline vehicles were calculated by Eq. (9). The control efficiency and the percentages of gasoline stations equipped with Stage-II systems are the two key factors influencing the final emissions."

(6) Line 244, 264, why China 4 LDGVs' EFs could be used for all non-motorcycle vehicle types and control technologies?

**Response:** The EFs were assumed to be the same for China 1 to China. The following description was added.

**Author's changes in manuscript:**

The evaporative emission control was keeping the same until China 6. Thus, there's no progress on emission reduction since China 1 to China 5 on evaporation. So, the emission factors of China 4 LDGVs could be used for all LDGVs. For the other vehicle types, no data is available from tests and the same EFs with LDGV were used.

(7) Eqs. (7), (9), (11), (12). The authors claimed that the units of EFs are g/hour. The reviewer believes that this is not correct.

**Response:** The equations and the explanation were revised. All the EFs including units were listed in the new Table S6.
**Author's changes in manuscript:**

**Table S6. Evaporation emission factors used in this study.**

|  |  | Parking duration | Unite | Emission factors |
|---|---|---|---|---|
| vehicles | Diurnal | <24 hour | g/hour | 0.094[1] |
|  |  | 24-48 hour | g/hour | 0.247[1] |
|  |  | >48 hour | g/hour | 0.339[1] |
|  | Hot soak |  | g/hour | 0.083[1] |
|  | Refueling (without control) |  | g/L | 0.848[1] |
|  | Running loss |  | g/hour | 11.6[2] |
| motorcycle |  |  | g/km | 0.57[3] |

Line 290. Is the motor gasoline consumption by province calculated or derived from official statistics? Methods or data sources should be provided.

**Response:** The gasoline consumption is from statistic data. A sentence was added to provide the method and data source.

**Author's changes in manuscript:**

"$CF_i$ represents the annual motor gasoline consumption of province i (L·year-1), which was retrieved from official statistics (China Energy Statistical Yearbook 2016) and 85% of total gasoline was assumed to be used in on-road vehicles."

Main text after Sect. 3.2 may need to be polished to make it read like a scientific article.

**Response:** Two native speakers polished the language of the paper. We also contacted Copernicus Publication copy-editing service. After this manuscript was accepted by ACP, they will polish the language.

The authors are suggested to check citations carefully before submitting the revised manuscript. Examples are: Line 61, change "Cai et al" to "Cai and Xie". Remove "(Cai et al., 2009)" in line 62 Yang et al., 2015 is mentioned several times in the manuscript (e.g., lines 95, 106, 150, 194, 205, 281, etc.). However, there are two references by Yang et al. in 2015. Letters a and b should be added to the year both in the in-text citation as well as in the reference list. Line 163-164, 179-180, "Zhao et al." to "Zhao et al. (2015, 2016)" and remove "(Zhao et al.; 2016; Zhao et al; 2015)" Line 275, 307, "ICCT, 2012" is not in the reference list Line 301, "MOVES, 2010" is not in the reference list Line 326, "Man et al., 2016" is not in the reference list. In the reference list, there are lots of references that are not cited in the main text. Please have them carefully checked before submitting the revised manuscript.

**Response:** We have checked all the citations. The reference list is match with those cited in the main text now. The Endnote templates from ACP website were used to format all the references. All the problems mentioned above were corrected in this revision.

Minor editorial issues:

Line 121, remove "five". According the introduction section, it seems that there are six deficiencies, while in Sect. 4, it seems the authors discussed four aspects.

**Response:** Accepted.

Line 217, "POA" should be defined in the first appearance.

**Response:** Accepted.

**Author's changes in manuscript:** "This ratio was similar to the VOCs or primary

organic aerosol (POA) emission ratios of heavy/light for trucks."

Line 257, "GTs"??

**Response:** Corrected.

Line 324, incomplete sentence

**Response:** Corrected.

Line 394, "eg." to "e.g., "

**Response:** Corrected.

What is the unit of EFs in Table S3?

**Response:** Added. The unit is mg/kg-fuel.

The caption of Figure 1 should be self-explained.

**Response:** Corrected.

**Author's changes in manuscript:** The caption was revised to "The percentages by vehicle types, fuel types and emission levels of China vehicle fleet".

There are grammatical errors throughout the manuscript. I strongly suggest a grammar checking by native English speaker before submitting the revised manuscript. Examples in the first five pages are: Abstract should be written in the present tense.

**Response:** A native speaker polished the language of the paper.

Line 41-42. Line 47, remove "the year of"

**Response:** Accepted.

Line 62, add "during" after "China"

**Response:** Accepted.

Line 63, "include" to "included", add "a" before "part"

**Response:** Accepted.

Line 68, "provide" o "provided"

**Response:** Accepted.

Line 70 remove "trend"

**Response:** Accepted.

Line 74, "has" to "have", "a non-ignorable contributor" to "non-ignorable contributors"

**Response:** Accepted.

Line 76 Line 81, "profile" to "profiles"

**Response:** Accepted.

Line 82, "with" to "to"

**Response:** Accepted.

Line 83, "were" to "are", "method section" to "Sect. 2"

**Response:** Accepted.

Line 84, "impact" to "impacts", "atmospheric condition" to "air quality"???

**Response:** Accepted.

Line 86, "complicate" to "complicated", add "of" after "a series"

**Response:** Accepted.

Line 90, "measurements" to "measurement", "none of the" to "to our knowledge, there is no", add "for China" at the end of this sentence

**Response:** Accepted.

Line 98 Line 100, "method" to "methods"

**Response:** Accepted.

Line 106, "emission" to "emissions were"

**Response:** Accepted.

Line 109, "common-used" to "commonly-used"

**Response:** Accepted.

Line 111, "provided" to "provide", "level" to "levels"

**Response:** Accepted.

Line 113, "recently" to "recent".

**Response:** Accepted.

Line 114, add "furthermore," at the beginning of the sentence, "provides" to "provide", "types" to "type"

**Response:** Accepted.

Line 115, remove "However,"

**Response:** Accepted.

Line 116-117, change to "More detailed vehicle population data by fuel type and by control technology are required to calculate emissions because they have been reported to . . ."

**Response:** Accepted.

Line 120, "were" to "are"

**Response:** Accepted.

Line 121, "were" to "are"

**Response:** Accepted.

Line 123, "were" to "are"

**Response:** Accepted.

Line 124, change to "there is no local IVOC emission factor reported"

**Response:** Accepted.

---

## Author Comment (AC2) · 30 Aug 2017

4    Anonymous Referee #2

6    General Comments: This work developed an updated speciated emission inventory of
7    VOCs and IVOCs from vehicles in China for the year of 2015 based on a set of state-
8    of-the-art methodologies and a mass of local measurement data. The strength of this
9    inventory is that massive GPS records and questionnaire analysis are collected to better
10   characterize the activity level. In addition, in terms of the method, this work improved
11   the emission estimation by including evaporative emission calculation and applying
12   road emission intensity based approach. This well-written and well-structured paper is
13   potentially important and will be valuable in the future for modelling the formation of
14   fine particles and ozone pollution in China. There are a few comments that need to be
15   addressed to improve the paper and make it more accessible to the wide audience of
16   users of the information presented.

17   **Response:** Thank you for the comments. We try our best to improve the manuscript
18   based on your comments. The point-by-point response is provided.

19   Specific Comments:

20   In the first place, the information need to be made available, for example through the
21   journal with a doi, or through the website of the author's institute.

22   **Response:** Accepted. Firstly, instead of providing figures and percentages, we have
23   revised and added several tables to provide the raw data and the emission data. Due to
24   the length limitation, the additional dataset are available upon request.

25   **Author's changes in manuscript:**

26   Table 1 to table 5 and table S1 to table S9 were added to provide information as detail
27   as possible.
28   Table 1 Population of different types of vehicles in China in 2015
29

A second recommendation is that speciated emission inventory of VOCs and IVOCs based on prevailing lumped chemical mechanisms like CB05 and SAPRC are suggested to be provided since that this emission database will be mainly used in chemical transport models.

**Response:** Accepted. A table and discussions were added.

**Author's changes in manuscript:**

Table S8. Assignments from Real Compounds to Carbon Bond 05 (CB05) Model Species for diesel exhaust, gasoline exhaust and evaporation in China (Gmol).

|        | Diesel exhaust | Gasoline exhaust | Evaporation |
|--------|----------------|------------------|-------------|
| PAR    | 7.179          | 39.017           | 72.452      |
| OLE    | 0.371          | 0.994            | 1.380       |
| TOL    | 0.217          | 2.389            | 0.507       |
| XYL    | 0.222          | 1.035            | 0.189       |
| FORM   | 4.425          | 2.700            | 0.215       |
| ALD2   | 1.219          | 1.071            | 0.095       |
| ETH    | 0.837          | N.D.             | 0.017       |
| ISOP   | N.D.           | N.D.             | 0.012       |
| MEOH   | N.D.           | N.D.             | N.D.        |
| ETOH   | N.D.           | N.D.             | N.D.        |
| ETHA   | N.D.           | 0.882            | 0.158       |
| IOLE   | N.D.           | N.D.             | 2.046       |
| ALDX   | 0.6852         | 1.309            | 0.128       |
| TERP   | N.D.           | N.D.             | N.D.        |
| UNR    | 1.773          | 8.276            | 5.762       |

There still large uncertainty lies in activity level, emission factor and the estimation method itself. Another recommendation is that uncertainty analysis ought to be conducted and more quantitative results should be provided in Section 3.3.

**Response:** Accepted.

**Author's changes in manuscript:**

The uncertainty for emission inventory is assessed using a Monte Carlo method. The probability distributions of key model parameters were established with our experimental data, investigation data and literature review (Table S7). Using these assumptions, a Monte Carlo model was run 10000 times to produce the estimate.

Inevitable uncertainties are present in VOCs emission inventories due to the use of different input data, including activity characteristics, emission factors and VOCs emission profiles. Total vehicle emissions of VOCs are 4.21 Tg yr−1 with a 95% confidence interval ranges from 2.90-6.54 Tg. The overall uncertainties in this inventory are estimated at −28.53 to 61.35% for total VOC emissions. The uncertainties of detailed categories were listed in Table S9. These confidence ranges are comparable to other bottom-up emission inventories.

**Table S7. Characteristics of probability distribution functions for selected key model parameters and input variables included in the uncertainty analysis.**

| Parameter or variable | | | Distribution | Standard division | The 95% confidence interval | | |
|---|---|---|---|---|---|---|---|
| | | | | | 2.5% percentile | 50 % percentile | 97.5% percentile |
| Evaporative emission factors | Diurnal emissions (g/hour) | 1-24hour | Log-Normal | 0.065 | 0.023 | 0.077 | 0.264 |
| | | 24-48hour | Log-Normal | 0.100 | 0.107 | 0.229 | 0.493 |
| | | >48hour | Log-Normal | 0.085 | 0.204 | 0.331 | 0.536 |
| | Hot Soak (g/hour) | | Log-Normal | 0.014 | 0.059 | 0.082 | 0.114 |
| | Base Refuelling (g/L) | | Log-Normal | 0.077 | 0.707 | 0.843 | 1.009 |
| | Running loss (g/hour) | | Log-Normal | 4.689 | 5.072 | 10.712 | 22.938 |
| | GMs (g/Km) | | Log-Normal | 0.550 | 0.086 | 0.415 | 1.945 |
| Parking duration per day in Beijing (hour) | | | Extreme Value | 1.1365 | 19.4652 | 22.3486 | 23.8540 |
| Parking duration per day in other provinces (hour) | | | Extreme Value | 0.9919 | 19.7238 | 22.2438 | 23.5538 |
| Percentage of parking events in Beijing | 0-1hour | | Log-Normal | 0.100 | 0.320 | 0.475 | 0.712 |
| | 1-24hour | | Log-Normal | 0.099 | 0.306 | 0.460 | 0.688 |
| | 24-48hour | | Log-Normal | 0.006 | 0.018 | 0.027 | 0.041 |
| | 48-72hour | | Log-Normal | 0.002 | 0.004 | 0.007 | 0.012 |
| | 72-119.5hour | | Log-Normal | 0.002 | 0.002 | 0.005 | 0.010 |
| | >119.5hour | | Log-Normal | 0.000 | 0.004 | 0.004 | 0.005 |
| Percentage of parking | 0-1hour | | Log-Normal | 0.124 | 0.352 | 0.539 | 0.834 |
| | 1-24hour | | Log-Normal | 0.079 | 0.290 | 0.420 | 0.605 |

| | | | | | | | |
|---|---|---|---|---|---|---|---|
| events in other provinces | 24-48hour | | Log-Normal | 0.002 | 0.007 | 0.010 | 0.015 |
| | 48-72hour | | Log-Normal | 0.000 | 0.002 | 0.003 | 0.004 |
| | 72-119.5hour | | Log-Normal | 0.002 | 0.000 | 0.002 | 0.007 |
| | >119.5hour | | Log-Normal | 0.004 | 0.000 | 0.001 | 0.010 |
| Percentage of parking duration in Beijing | 0-1hour | | Log-Normal | 0.006 | 0.020 | 0.029 | 0.043 |
| | 1-24hour | | Log-Normal | 0.099 | 0.316 | 0.471 | 0.703 |
| | 24-48hour | | Log-Normal | 0.040 | 0.101 | 0.162 | 0.260 |
| | 48-72hour | | Log-Normal | 0.014 | 0.048 | 0.071 | 0.103 |
| | 72-119.5hour | | Log-Normal | 0.020 | 0.050 | 0.080 | 0.127 |
| | >119.5hour | | Log-Normal | 0.040 | 0.103 | 0.163 | 0.255 |
| Percentage of parking duration in other provinces | 0-1hour | | Log-Normal | 0.020 | 0.024 | 0.049 | 0.101 |
| | 1-24hour | | Log-Normal | 0.121 | 0.433 | 0.628 | 0.902 |
| | 24-48hour | | Log-Normal | 0.184 | 0.004 | 0.043 | 0.468 |
| | 48-72hour | | Log-Normal | 0.010 | 0.030 | 0.046 | 0.069 |
| | 72-119.5hour | | Log-Normal | 0.020 | 0.022 | 0.047 | 0.098 |
| | >119.5hour | | Log-Normal | 0.020 | 0.084 | 0.117 | 0.161 |
| Tailpipe Emission factors of passenger vehicles (g/Km) | GMs | | Log-Normal | 0.56 | 0.52 | 1.16 | 2.64 |
| | LDGTAs | China0 | Log-Normal | 1.694 | 1.550 | 3.519 | 8.045 |
| | | China1 | Log-Normal | 0.599 | 0.558 | 1.255 | 2.839 |
| | | China2 | Log-Normal | 0.418 | 0.392 | 0.891 | 1.968 |
| | | China3 | Log-Normal | 0.200 | 0.184 | 0.416 | 0.957 |
| | | China4 | Log-Normal | 0.121 | 0.112 | 0.254 | 0.582 |
| | | China5 | Log-Normal | 0.114 | 0.104 | 0.236 | 0.543 |
| | LDDTAs | China0 | Log-Normal | 0.337 | 0.311 | 0.726 | 1.608 |
| | | China1 | Log-Normal | 0.031 | 0.028 | 0.065 | 0.150 |
| | | China2 | Log-Normal | 0.020 | 0.019 | 0.042 | 0.096 |
| | | China3 | Log-Normal | 0.010 | 0.010 | 0.022 | 0.050 |
| | | China4 | Log-Normal | 0.007 | 0.006 | 0.015 | 0.033 |
| | | China5 | Log-Normal | 0.007 | 0.006 | 0.015 | 0.033 |
| | LDATAs | China0 | Log-Normal | 1.649 | 1.516 | 3.471 | 7.828 |
| | | China1 | Log-Normal | 0.187 | 0.174 | 0.396 | 0.884 |
| | | China2 | Log-Normal | 0.176 | 0.159 | 0.367 | 0.829 |
| | | China3 | Log-Normal | 0.050 | 0.046 | 0.105 | 0.241 |
| | | China4 | Log-Normal | 0.029 | 0.027 | 0.060 | 0.139 |
| | | China5 | Log-Normal | 0.127 | 0.118 | 0.270 | 0.608 |
| | LDGPVs | China0 | Log-Normal | 1.181 | 1.105 | 2.473 | 5.687 |
| | | China1 | Log-Normal | 0.293 | 0.269 | 0.608 | 1.385 |
| | | China2 | Log-Normal | 0.140 | 0.127 | 0.287 | 0.654 |
| | | China3 | Log-Normal | 0.083 | 0.077 | 0.174 | 0.395 |
| | | China4 | Log-Normal | 0.033 | 0.031 | 0.069 | 0.158 |

| | | | | | | |
|---|---|---|---|---|---|---|
| | China5 | Log-Normal | 0.025 | 0.023 | 0.052 | 0.115 |
| LDDPVs | China0 | Log-Normal | 0.337 | 0.311 | 0.726 | 1.608 |
| | China1 | Log-Normal | 0.031 | 0.028 | 0.065 | 0.150 |
| | China2 | Log-Normal | 0.020 | 0.019 | 0.042 | 0.096 |
| | China3 | Log-Normal | 0.010 | 0.010 | 0.022 | 0.050 |
| | China4 | Log-Normal | 0.007 | 0.006 | 0.015 | 0.033 |
| | China5 | Log-Normal | 0.007 | 0.006 | 0.015 | 0.033 |
| LDAPVs | China0 | Log-Normal | 0.977 | 0.900 | 2.071 | 4.581 |
| | China1 | Log-Normal | 0.104 | 0.095 | 0.217 | 0.486 |
| | China2 | Log-Normal | 0.073 | 0.067 | 0.151 | 0.347 |
| | China3 | Log-Normal | 0.065 | 0.023 | 0.077 | 0.264 |
| | China4 | Log-Normal | 0.027 | 0.025 | 0.056 | 0.127 |
| | China5 | Log-Normal | 0.039 | 0.037 | 0.084 | 0.186 |
| MDGBUs | China0 | Log-Normal | 2.223 | 2.100 | 4.741 | 10.559 |
| | China1 | Log-Normal | 2.306 | 2.142 | 4.859 | 10.957 |
| | China2 | Log-Normal | 0.852 | 0.788 | 1.805 | 4.030 |
| | China3 | Log-Normal | 0.383 | 0.346 | 0.791 | 1.838 |
| | China4 | Log-Normal | 0.184 | 0.167 | 0.380 | 0.872 |
| | China5 | Log-Normal | 0.184 | 0.167 | 0.380 | 0.872 |
| MDDBUs | China0 | Log-Normal | 1.184 | 1.068 | 2.424 | 5.614 |
| | China1 | Log-Normal | 0.254 | 0.234 | 0.532 | 1.211 |
| | China2 | Log-Normal | 0.153 | 0.141 | 0.323 | 0.728 |
| | China3 | Log-Normal | 0.122 | 0.114 | 0.260 | 0.583 |
| | China4 | Log-Normal | 0.047 | 0.042 | 0.098 | 0.220 |
| | China5 | Log-Normal | 0.024 | 0.021 | 0.050 | 0.112 |
| MDABUs | China0 | Log-Normal | 1.694 | 1.550 | 3.519 | 8.045 |
| | China1 | Log-Normal | 1.415 | 1.299 | 2.945 | 6.765 |
| | China2 | Log-Normal | 1.256 | 1.146 | 2.609 | 5.943 |
| | China3 | Log-Normal | 0.745 | 0.697 | 1.555 | 3.567 |
| | China4 | Log-Normal | 0.517 | 0.484 | 1.100 | 2.460 |
| | China5 | Log-Normal | 0.517 | 0.484 | 1.100 | 2.460 |
| MDGPVs | China0 | Log-Normal | 1.623 | 1.482 | 3.364 | 7.725 |
| | China1 | Log-Normal | 1.123 | 1.043 | 2.351 | 5.301 |
| | China2 | Log-Normal | 0.628 | 0.587 | 1.324 | 2.993 |
| | China3 | Log-Normal | 0.165 | 0.150 | 0.338 | 0.779 |
| | China4 | Log-Normal | 0.047 | 0.042 | 0.098 | 0.220 |
| | China5 | Log-Normal | 0.047 | 0.042 | 0.098 | 0.220 |
| MDDPVs | China0 | Log-Normal | 0.663 | 0.602 | 1.373 | 3.165 |
| | China1 | Log-Normal | 0.612 | 0.576 | 1.301 | 2.933 |
| | China2 | Log-Normal | 0.185 | 0.172 | 0.391 | 0.873 |

|  | China3 | Log-Normal | 0.160 | 0.146 | 0.335 | 0.764 |
|  | China4 | Log-Normal | 0.166 | 0.155 | 0.348 | 0.797 |
|  | China5 | Log-Normal | 0.166 | 0.155 | 0.348 | 0.797 |
| MDAPVs | China0 | Log-Normal | 0.834 | 0.763 | 1.753 | 4.002 |
|  | China1 | Log-Normal | 0.704 | 0.650 | 1.479 | 3.335 |
|  | China2 | Log-Normal | 0.628 | 0.571 | 1.305 | 2.985 |
|  | China3 | Log-Normal | 0.376 | 0.347 | 0.780 | 1.797 |
|  | China4 | Log-Normal | 0.259 | 0.246 | 0.550 | 1.245 |
|  | China5 | Log-Normal | 0.259 | 0.246 | 0.550 | 1.245 |
| MDGBUs | China0 | Log-Normal | 2.223 | 2.100 | 4.741 | 10.559 |
|  | China1 | Log-Normal | 2.306 | 2.142 | 4.859 | 10.957 |
|  | China2 | Log-Normal | 0.852 | 0.788 | 1.805 | 4.030 |
|  | China3 | Log-Normal | 0.383 | 0.346 | 0.791 | 1.838 |
|  | China4 | Log-Normal | 0.184 | 0.167 | 0.380 | 0.872 |
|  | China5 | Log-Normal | 0.184 | 0.167 | 0.380 | 0.872 |
| HDDBUs | China0 | Log-Normal | 1.184 | 1.068 | 2.424 | 5.614 |
|  | China1 | Log-Normal | 0.254 | 0.234 | 0.532 | 1.211 |
|  | China2 | Log-Normal | 0.153 | 0.141 | 0.323 | 0.728 |
|  | China3 | Log-Normal | 0.122 | 0.114 | 0.260 | 0.583 |
|  | China4 | Log-Normal | 0.047 | 0.042 | 0.098 | 0.220 |
|  | China5 | Log-Normal | 0.024 | 0.021 | 0.050 | 0.112 |
| HDABUs | China0 | Log-Normal | 1.694 | 1.550 | 3.519 | 8.045 |
|  | China1 | Log-Normal | 1.415 | 1.299 | 2.945 | 6.765 |
|  | China2 | Log-Normal | 1.256 | 1.146 | 2.609 | 5.943 |
|  | China3 | Log-Normal | 0.745 | 0.697 | 1.555 | 3.567 |
|  | China4 | Log-Normal | 0.517 | 0.484 | 1.100 | 2.460 |
|  | China5 | Log-Normal | 0.517 | 0.484 | 1.100 | 2.460 |
| HDGPVs | China0 | Log-Normal | 2.223 | 2.100 | 4.741 | 10.559 |
|  | China1 | Log-Normal | 2.306 | 2.142 | 4.859 | 10.957 |
|  | China2 | Log-Normal | 0.852 | 0.788 | 1.805 | 4.030 |
|  | China3 | Log-Normal | 0.383 | 0.346 | 0.791 | 1.838 |
|  | China4 | Log-Normal | 0.184 | 0.167 | 0.380 | 0.872 |
|  | China5 | Log-Normal | 0.184 | 0.167 | 0.380 | 0.872 |
| HDDPVs | China0 | Log-Normal | 1.184 | 1.068 | 2.424 | 5.614 |
|  | China1 | Log-Normal | 0.254 | 0.234 | 0.532 | 1.211 |
|  | China2 | Log-Normal | 0.153 | 0.141 | 0.323 | 0.728 |
|  | China3 | Log-Normal | 0.122 | 0.114 | 0.260 | 0.583 |
|  | China4 | Log-Normal | 0.047 | 0.042 | 0.098 | 0.220 |
|  | China5 | Log-Normal | 0.024 | 0.021 | 0.050 | 0.112 |
| HDAPVs | China0 | Log-Normal | 1.694 | 1.550 | 3.519 | 8.045 |

| | | | | | | | |
|---|---|---|---|---|---|---|---|
| | | China1 | Log-Normal | 1.415 | 1.299 | 2.945 | 6.765 |
| | | China2 | Log-Normal | 1.256 | 1.146 | 2.609 | 5.943 |
| | | China3 | Log-Normal | 0.745 | 0.697 | 1.555 | 3.567 |
| | | China4 | Log-Normal | 0.517 | 0.484 | 1.100 | 2.460 |
| | | China5 | Log-Normal | 0.517 | 0.484 | 1.100 | 2.460 |
| VKT of passenger vehicles in Beijing (Km) | LDGTAs | China0 | Log-Normal | 78550 | 38951 | 113204 | 330220 |
| | LDDTAs | China0 | Log-Normal | 78550 | 38951 | 113204 | 330220 |
| | LDATAs | China0 | Log-Normal | 78550 | 38951 | 113204 | 330220 |
| | LDGPVs | China0 | Log-Normal | 7841 | 3973 | 11362 | 33524 |
| | LDDPVs | China0 | Log-Normal | 7841 | 3973 | 11362 | 33524 |
| | LDAPVs | China0 | Log-Normal | 7841 | 3973 | 11362 | 33524 |
| | MDGBUs | China0 | Log-Normal | 4991 | 40260 | 50093 | 59910 |
| | MDDBUs | China0 | Log-Normal | 4991 | 40260 | 50093 | 59910 |
| | MDABUs | China0 | Log-Normal | 4991 | 40260 | 50093 | 59910 |
| | MDGPVs | China0 | Log-Normal | 3143 | 25009 | 31310 | 37380 |
| | MDDPVs | China0 | Log-Normal | 3143 | 25009 | 31310 | 37380 |
| | MDAPVs | China0 | Log-Normal | 3143 | 25009 | 31310 | 37380 |
| | HDGBUs | China0 | Log-Normal | 4991 | 40260 | 50093 | 59910 |
| | HDDBUs | China0 | Log-Normal | 4991 | 40260 | 50093 | 59910 |
| | HDABUs | China0 | Log-Normal | 4991 | 40260 | 50093 | 59910 |
| | HDGPVs | China0 | Log-Normal | 11401 | 92557 | 114757 | 136940 |
| | HDDPVs | China0 | Log-Normal | 11401 | 92557 | 114757 | 136940 |
| | HDAPVs | China0 | Log-Normal | 11401 | 92557 | 114757 | 136940 |
| VKT of passenger vehicles in other provinces (Km) | LDGTAs | China0 | Log-Normal | 78325 | 43077 | 120437 | 342273 |
| | LDDTAs | China0 | Log-Normal | 78325 | 43077 | 120437 | 342273 |
| | LDATAs | China0 | Log-Normal | 78325 | 43077 | 120437 | 342273 |
| | LDGPVs | China0 | Log-Normal | 10796 | 6013 | 16571 | 46419 |
| | LDDPVs | China0 | Log-Normal | 10796 | 6013 | 16571 | 46419 |
| | LDAPVs | China0 | Log-Normal | 10796 | 6013 | 16571 | 46419 |
| | MDGBUs | China0 | Log-Normal | 4991 | 40260 | 50093 | 59910 |
| | MDDBUs | China0 | Log-Normal | 4991 | 40260 | 50093 | 59910 |
| | MDABUs | China0 | Log-Normal | 4991 | 40260 | 50093 | 59910 |
| | MDGPVs | China0 | Log-Normal | 3143 | 25009 | 31310 | 37380 |
| | MDDPVs | China0 | Log-Normal | 3143 | 25009 | 31310 | 37380 |
| | MDAPVs | China0 | Log-Normal | 3143 | 25009 | 31310 | 37380 |
| | HDGBUs | China0 | Log-Normal | 4991 | 40260 | 50093 | 59910 |
| | HDDBUs | China0 | Log-Normal | 4991 | 40260 | 50093 | 59910 |
| | HDABUs | China0 | Log-Normal | 4991 | 40260 | 50093 | 59910 |
| | HDGPVs | China0 | Log-Normal | 11401 | 92557 | 114757 | 136940 |
| | HDDPVs | China0 | Log-Normal | 11401 | 92557 | 114757 | 136940 |

| | | | | | | | |
|---|---|---|---|---|---|---|---|
| | HDAPVs | China0 | Log-Normal | 11401 | 92557 | 114757 | 136940 |
| Emission factors on Urban road (g/Km) | LDGTs | China0 | Normal | 2.331 | 2.172 | 4.895 | 10.942 |
| | | China1 | Normal | 1.604 | 1.454 | 3.301 | 7.527 |
| | | China2 | Normal | 1.049 | 0.962 | 2.195 | 5.010 |
| | | China3 | Normal | 0.279 | 0.259 | 0.583 | 1.323 |
| | | China4 | Normal | 0.078 | 0.070 | 0.160 | 0.367 |
| | | China5 | Normal | 0.078 | 0.070 | 0.160 | 0.367 |
| | MDGTs | China0 | Normal | 3.305 | 3.014 | 6.799 | 15.521 |
| | | China1 | Normal | 3.223 | 2.942 | 6.688 | 15.221 |
| | | China2 | Normal | 1.436 | 1.319 | 3.012 | 6.806 |
| | | China3 | Normal | 0.643 | 0.601 | 1.361 | 3.042 |
| | | China4 | Normal | 0.275 | 0.249 | 0.561 | 1.307 |
| | | China5 | Normal | 0.275 | 0.249 | 0.561 | 1.307 |
| | HDGTs | China0 | Normal | 3.200 | 2.955 | 6.680 | 15.063 |
| | | China1 | Normal | 3.148 | 2.944 | 6.678 | 14.911 |
| | | China2 | Normal | 1.393 | 1.346 | 2.989 | 6.689 |
| | | China3 | Normal | 0.639 | 0.588 | 1.333 | 3.062 |
| | | China4 | Normal | 0.261 | 0.245 | 0.550 | 1.242 |
| | | China5 | Normal | 0.259 | 0.242 | 0.549 | 1.237 |
| | LDGTs | China0 | Normal | 1.774 | 1.657 | 3.728 | 8.432 |
| | | China1 | Normal | 1.182 | 1.107 | 2.497 | 5.675 |
| | | China2 | Normal | 0.802 | 0.754 | 1.699 | 3.831 |
| | | China3 | Normal | 0.233 | 0.218 | 0.481 | 1.120 |
| | | China4 | Normal | 0.064 | 0.059 | 0.134 | 0.302 |
| | | China5 | Normal | 0.064 | 0.059 | 0.134 | 0.302 |
| Emission factors on Provincial road (g/Km) | MDGTs | China0 | Normal | 0.559 | 4.572 | 5.555 | 6.775 |
| | | China1 | Normal | 2.399 | 2.225 | 5.036 | 11.250 |
| | | China2 | Normal | 1.053 | 0.997 | 2.232 | 5.023 |
| | | China3 | Normal | 0.486 | 0.453 | 1.025 | 2.268 |
| | | China4 | Normal | 0.200 | 0.190 | 0.422 | 0.962 |
| | | China5 | Normal | 0.200 | 0.190 | 0.422 | 0.962 |
| | HDGTs | China0 | Normal | 2.402 | 2.198 | 4.985 | 11.389 |
| | | China1 | Normal | 2.359 | 2.185 | 5.015 | 11.257 |
| | | China2 | Normal | 1.099 | 0.983 | 2.252 | 5.223 |
| | | China3 | Normal | 0.470 | 0.446 | 0.997 | 2.254 |
| | | China4 | Normal | 0.045 | 0.368 | 0.447 | 0.543 |
| | | China5 | Normal | 0.202 | 0.181 | 0.414 | 0.947 |
| Emission factors on | LDGTs | China0 | Normal | 1.902 | 1.789 | 3.988 | 9.089 |
| | | China1 | Normal | 1.268 | 1.181 | 2.697 | 6.060 |
| | | China2 | Normal | 0.846 | 0.775 | 1.764 | 3.999 |

| | | | | | | | |
|---|---|---|---|---|---|---|---|
| National road (g/Km) | | China3 | Normal | 0.245 | 0.219 | 0.500 | 1.159 |
| | | China4 | Normal | 0.066 | 0.062 | 0.138 | 0.317 |
| | | China5 | Normal | 0.066 | 0.062 | 0.138 | 0.317 |
| | MDGTs | China0 | Normal | 2.612 | 2.461 | 5.547 | 12.615 |
| | | China1 | Normal | 2.630 | 2.403 | 5.400 | 12.396 |
| | | China2 | Normal | 1.150 | 1.070 | 2.424 | 5.444 |
| | | China3 | Normal | 0.521 | 0.482 | 1.100 | 2.456 |
| | | China4 | Normal | 0.218 | 0.201 | 0.461 | 1.031 |
| | | China5 | Normal | 0.218 | 0.201 | 0.461 | 1.031 |
| | HDGTs | China0 | Normal | 2.575 | 2.427 | 5.430 | 12.302 |
| | | China1 | Normal | 2.639 | 2.391 | 5.387 | 12.523 |
| | | China2 | Normal | 1.140 | 1.072 | 2.415 | 5.435 |
| | | China3 | Normal | 0.513 | 0.484 | 1.088 | 2.454 |
| | | China4 | Normal | 0.215 | 0.199 | 0.446 | 1.020 |
| | | China5 | Normal | 0.211 | 0.198 | 0.449 | 0.997 |
| Emission factors on Freeway (g/Km) | LDGTs | China0 | Normal | 1.801 | 1.691 | 3.760 | 8.437 |
| | | China1 | Normal | 1.219 | 1.121 | 2.542 | 5.812 |
| | | China2 | Normal | 0.808 | 0.746 | 1.688 | 3.817 |
| | | China3 | Normal | 0.237 | 0.214 | 0.487 | 1.117 |
| | | China4 | Normal | 0.065 | 0.059 | 0.136 | 0.309 |
| | | China5 | Normal | 0.065 | 0.059 | 0.136 | 0.309 |
| | MDGTs | China0 | Normal | 2.525 | 2.249 | 5.277 | 11.911 |
| | | China1 | Normal | 2.418 | 2.264 | 5.101 | 11.534 |
| | | China2 | Normal | 1.086 | 1.003 | 2.296 | 5.215 |
| | | China3 | Normal | 0.487 | 0.457 | 1.048 | 2.313 |
| | | China4 | Normal | 0.207 | 0.188 | 0.431 | 0.993 |
| | | China5 | Normal | 0.207 | 0.188 | 0.431 | 0.993 |
| | HDGTs | China0 | Normal | 2.410 | 2.242 | 5.132 | 11.470 |
| | | China1 | Normal | 2.430 | 2.327 | 5.150 | 11.495 |
| | | China2 | Normal | 1.089 | 0.992 | 2.282 | 5.249 |
| | | China3 | Normal | 0.485 | 0.456 | 1.029 | 2.318 |
| | | China4 | Normal | 0.203 | 0.182 | 0.420 | 0.968 |
| | | China5 | Normal | 0.198 | 0.187 | 0.418 | 0.941 |
| Emission factors on other type roads (g/Km) | LDGTs | China0 | Normal | 3.066 | 2.829 | 6.411 | 14.780 |
| | | China1 | Normal | 1.994 | 1.857 | 4.246 | 9.496 |
| | | China2 | Normal | 1.370 | 1.238 | 2.848 | 6.525 |
| | | China3 | Normal | 0.352 | 0.320 | 0.734 | 1.679 |
| | | China4 | Normal | 0.094 | 0.089 | 0.204 | 0.452 |
| | | China5 | Normal | 0.094 | 0.089 | 0.204 | 0.452 |
| | MDGTs | China0 | Normal | 4.173 | 3.949 | 8.844 | 20.062 |

| | | | | | | | |
|---|---|---|---|---|---|---|---|
| | | China1 | Normal | 4.122 | 3.823 | 8.777 | 19.875 |
| | | China2 | Normal | 1.900 | 1.707 | 3.882 | 9.052 |
| | | China3 | Normal | 0.839 | 0.781 | 1.759 | 3.984 |
| | | China4 | Normal | 0.357 | 0.325 | 0.738 | 1.709 |
| | | China5 | Normal | 0.357 | 0.325 | 0.738 | 1.709 |
| | HDGTs | China0 | Normal | 4.102 | 3.794 | 8.706 | 19.494 |
| | | China1 | Normal | 4.055 | 3.828 | 8.734 | 19.229 |
| | | China2 | Normal | 1.847 | 1.712 | 3.882 | 8.921 |
| | | China3 | Normal | 0.843 | 0.771 | 1.744 | 3.988 |
| | | China4 | Normal | 0.336 | 0.322 | 0.715 | 1.600 |
| | | China5 | Normal | 0.344 | 0.317 | 0.720 | 1.632 |
| Emission factors on urban road (g/Km) | LDDTs | China0 | Normal | 0.990 | 0.932 | 2.093 | 4.717 |
| | | China1 | Normal | 0.966 | 0.894 | 2.008 | 4.547 |
| | | China2 | Normal | 0.615 | 0.569 | 1.300 | 2.913 |
| | | China3 | Normal | 0.168 | 0.158 | 0.351 | 0.787 |
| | | China4 | Normal | 0.085 | 0.078 | 0.178 | 0.406 |
| | | China5 | Normal | 0.085 | 0.078 | 0.178 | 0.406 |
| | MDDTs | China0 | Normal | 2.125 | 1.942 | 4.434 | 10.110 |
| | | China1 | Normal | 0.772 | 0.701 | 1.596 | 3.656 |
| | | China2 | Normal | 0.200 | 0.185 | 0.414 | 0.940 |
| | | China3 | Normal | 0.096 | 0.089 | 0.199 | 0.457 |
| | | China4 | Normal | 0.048 | 0.045 | 0.103 | 0.233 |
| | | China5 | Normal | 0.048 | 0.045 | 0.103 | 0.233 |
| | HDDTs | China0 | Normal | 1.900 | 1.775 | 4.023 | 9.117 |
| | | China1 | Normal | 0.417 | 0.388 | 0.887 | 2.006 |
| | | China2 | Normal | 0.245 | 0.228 | 0.516 | 1.154 |
| | | China3 | Normal | 0.121 | 0.113 | 0.252 | 0.576 |
| | | China4 | Normal | 0.060 | 0.055 | 0.127 | 0.286 |
| | | China5 | Normal | 0.060 | 0.055 | 0.127 | 0.286 |
| Emission factors on provincial road (g/Km) | LDDTs | China0 | Normal | 0.745 | 0.668 | 1.552 | 3.518 |
| | | China1 | Normal | 0.719 | 0.667 | 1.515 | 3.409 |
| | | China2 | Normal | 0.487 | 0.444 | 0.998 | 2.317 |
| | | China3 | Normal | 0.138 | 0.130 | 0.294 | 0.658 |
| | | China4 | Normal | 0.072 | 0.065 | 0.148 | 0.339 |
| | | China5 | Normal | 0.072 | 0.065 | 0.148 | 0.339 |
| | MDDTs | China0 | Normal | 1.596 | 1.477 | 3.328 | 7.576 |
| | | China1 | Normal | 0.575 | 0.521 | 1.211 | 2.727 |
| | | China2 | Normal | 0.149 | 0.135 | 0.311 | 0.704 |
| | | China3 | Normal | 0.072 | 0.067 | 0.151 | 0.343 |
| | | China4 | Normal | 0.036 | 0.034 | 0.077 | 0.173 |

| | | | | | | | |
|---|---|---|---|---|---|---|---|
| | | China5 | Normal | 0.036 | 0.034 | 0.077 | 0.173 |
| | HDDTs | China0 | Normal | 1.465 | 1.322 | 3.030 | 6.879 |
| | | China1 | Normal | 0.319 | 0.290 | 0.665 | 1.514 |
| | | China2 | Normal | 0.188 | 0.170 | 0.385 | 0.890 |
| | | China3 | Normal | 0.090 | 0.083 | 0.190 | 0.424 |
| | | China4 | Normal | 0.046 | 0.042 | 0.096 | 0.217 |
| | | China5 | Normal | 0.046 | 0.042 | 0.096 | 0.217 |
| Emission factors on national road (g/Km) | LDDTs | China0 | Normal | 0.810 | 0.736 | 1.698 | 3.816 |
| | | China1 | Normal | 0.765 | 0.722 | 1.644 | 3.692 |
| | | China2 | Normal | 0.494 | 0.454 | 1.034 | 2.351 |
| | | China3 | Normal | 0.143 | 0.134 | 0.300 | 0.693 |
| | | China4 | Normal | 0.074 | 0.067 | 0.154 | 0.352 |
| | | China5 | Normal | 0.074 | 0.067 | 0.154 | 0.352 |
| | MDDTs | China0 | Normal | 1.730 | 1.585 | 3.640 | 8.182 |
| | | China1 | Normal | 0.611 | 0.571 | 1.299 | 2.923 |
| | | China2 | Normal | 0.161 | 0.150 | 0.337 | 0.764 |
| | | China3 | Normal | 0.078 | 0.071 | 0.162 | 0.369 |
| | | China4 | Normal | 0.040 | 0.036 | 0.083 | 0.190 |
| | | China5 | Normal | 0.040 | 0.036 | 0.083 | 0.190 |
| | HDDTs | China0 | Normal | 1.570 | 1.431 | 3.270 | 7.502 |
| | | China1 | Normal | 0.345 | 0.321 | 0.722 | 1.635 |
| | | China2 | Normal | 0.202 | 0.183 | 0.419 | 0.953 |
| | | China3 | Normal | 0.099 | 0.092 | 0.205 | 0.465 |
| | | China4 | Normal | 0.050 | 0.046 | 0.104 | 0.236 |
| | | China5 | Normal | 0.050 | 0.046 | 0.104 | 0.236 |
| Emission factors on freeway (g/Km) | LDDTs | China0 | Normal | 0.757 | 0.696 | 1.593 | 3.582 |
| | | China1 | Normal | 0.715 | 0.699 | 1.555 | 3.487 |
| | | China2 | Normal | 0.480 | 0.433 | 0.983 | 2.249 |
| | | China3 | Normal | 0.141 | 0.132 | 0.297 | 0.677 |
| | | China4 | Normal | 0.071 | 0.066 | 0.148 | 0.336 |
| | | China5 | Normal | 0.071 | 0.066 | 0.148 | 0.336 |
| | MDDTs | China0 | Normal | 1.639 | 1.506 | 3.402 | 7.777 |
| | | China1 | Normal | 0.582 | 0.529 | 1.221 | 2.748 |
| | | China2 | Normal | 0.153 | 0.141 | 0.322 | 0.724 |
| | | China3 | Normal | 0.073 | 0.067 | 0.154 | 0.345 |
| | | China4 | Normal | 0.037 | 0.034 | 0.078 | 0.175 |
| | | China5 | Normal | 0.037 | 0.034 | 0.078 | 0.175 |
| | HDDTs | China0 | Normal | 1.456 | 1.328 | 3.075 | 6.925 |
| | | China1 | Normal | 0.324 | 0.301 | 0.682 | 1.522 |
| | | China2 | Normal | 0.187 | 0.175 | 0.394 | 0.892 |

| | | | | | | | |
|---|---|---|---|---|---|---|---|
| | | China3 | Normal | 0.021 | 0.173 | 0.210 | 0.255 |
| | | China4 | Normal | 0.046 | 0.044 | 0.097 | 0.219 |
| | | China5 | Normal | 0.046 | 0.044 | 0.097 | 0.219 |
| Emission factors on other type roads (g/Km) | MDDTs | China0 | Normal | 1.286 | 1.182 | 2.694 | 6.171 |
| | | China1 | Normal | 1.214 | 1.176 | 2.635 | 5.828 |
| | | China2 | Normal | 0.793 | 0.732 | 1.667 | 3.801 |
| | | China3 | Normal | 0.208 | 0.192 | 0.439 | 0.992 |
| | | China4 | Normal | 0.105 | 0.100 | 0.224 | 0.506 |
| | | China5 | Normal | 0.105 | 0.100 | 0.224 | 0.506 |
| | | China0 | Normal | 2.701 | 2.537 | 5.734 | 12.858 |
| | | China1 | Normal | 0.972 | 0.928 | 2.080 | 4.628 |
| | MDDTs | China2 | Normal | 0.259 | 0.238 | 0.546 | 1.232 |
| | | China3 | Normal | 0.126 | 0.115 | 0.260 | 0.594 |
| | | China4 | Normal | 0.063 | 0.057 | 0.134 | 0.296 |
| | | China5 | Normal | 0.063 | 0.057 | 0.134 | 0.296 |
| | HDDTs | China0 | Normal | 2.497 | 2.292 | 5.314 | 11.801 |
| | | China1 | Normal | 0.561 | 0.517 | 1.170 | 2.649 |
| | | China2 | Normal | 0.320 | 0.299 | 0.675 | 1.527 |
| | | China3 | Normal | 0.156 | 0.149 | 0.328 | 0.749 |
| | | China4 | Normal | 0.079 | 0.072 | 0.165 | 0.374 |
| | | China5 | Normal | 0.079 | 0.072 | 0.165 | 0.374 |
| VKT of freight trucks (Km) | LDGTs | China0 | Normal | 2231 | 17804 | 22144 | 26459 |
| | | China1 | Normal | 2231 | 17804 | 22144 | 26459 |
| | | China2 | Normal | 2613 | 21212 | 26282 | 31510 |
| | | China3 | Normal | 2962 | 23701 | 29470 | 35296 |
| | | China4 | Normal | 3412 | 27517 | 34137 | 40880 |
| | | China5 | Normal | 3412 | 27517 | 34137 | 40880 |
| | LDDTs | China0 | Normal | 1936 | 15457 | 19261 | 23046 |
| | | China1 | Normal | 1936 | 15457 | 19261 | 23046 |
| | | China2 | Normal | 2698 | 21680 | 26977 | 32181 |
| | | China3 | Normal | 3642 | 29375 | 36624 | 43686 |
| | | China4 | Normal | 4564 | 36264 | 45280 | 54401 |
| | | China5 | Normal | 4564 | 36264 | 45280 | 54401 |
| | MDGTs | China0 | Normal | 3523 | 28361 | 35216 | 42199 |
| | | China1 | Normal | 3523 | 28361 | 35216 | 42199 |
| | | China2 | Normal | 4003 | 32747 | 40789 | 48541 |
| | | China3 | Normal | 4801 | 38505 | 47870 | 57236 |
| | | China4 | Normal | 5273 | 43193 | 53491 | 63709 |
| | | China5 | Normal | 5273 | 43193 | 53491 | 63709 |
| | MDDTs | China0 | Normal | 2126 | 17035 | 21228 | 25407 |

| | | | | | | | |
|---|---|---|---|---|---|---|---|
| | | China1 | Normal | 2126 | 17035 | 21228 | 25407 |
| | | China2 | Normal | 2833 | 22651 | 28170 | 33692 |
| | | China3 | Normal | 3616 | 29244 | 36345 | 43352 |
| | | China4 | Normal | 5998 | 48661 | 60346 | 72179 |
| | | China5 | Normal | 5998 | 48661 | 60346 | 72179 |
| | HDGTs | China0 | Normal | 2747 | 22236 | 27753 | 33098 |
| | | China1 | Normal | 2747 | 22236 | 27753 | 33098 |
| | | China2 | Normal | 3343 | 26601 | 33215 | 39661 |
| | | China3 | Normal | 4031 | 32340 | 40265 | 48241 |
| | | China4 | Normal | 4524 | 36913 | 45806 | 54752 |
| | | China5 | Normal | 4524 | 36913 | 45806 | 54752 |
| | HDDTs | China0 | Normal | 2430 | 19566 | 24330 | 29119 |
| | | China1 | Normal | 2430 | 19566 | 24330 | 29119 |
| | | China2 | Normal | 3814 | 31028 | 38551 | 46017 |
| | | China3 | Normal | 6316 | 51858 | 64083 | 76300 |
| | | China4 | Normal | 9863 | 78952 | 98396 | 117258 |
| | | China5 | Normal | 9863 | 78952 | 98396 | 117258 |
| Percentage of driving distance on different type roads | MDG(D)Ts | Urban road | Normal | 0.010 | 0.219 | 0.239 | 0.258 |
| | | Provincial road | Normal | 0.020 | 0.303 | 0.342 | 0.381 |
| | | National road | Normal | 0.007 | 0.124 | 0.137 | 0.151 |
| | | Freeway | Normal | 0.014 | 0.247 | 0.274 | 0.301 |
| | | others | Normal | 0.000 | 0.007 | 0.008 | 0.009 |
| | LDG(D)Ts | Urban road | Normal | 0.020 | 0.314 | 0.353 | 0.392 |
| | | Provincial road | Normal | 0.009 | 0.167 | 0.185 | 0.203 |
| | | National road | Normal | 0.011 | 0.196 | 0.218 | 0.239 |
| | | Freeway | Normal | 0.010 | 0.195 | 0.215 | 0.234 |
| | | others | Normal | 0.001 | 0.028 | 0.031 | 0.034 |
| | HDG(D)Ts | Urban road | Normal | 0.007 | 0.128 | 0.142 | 0.155 |
| | | Provincial road | Normal | 0.010 | 0.173 | 0.192 | 0.212 |
| | | National road | Normal | 0.010 | 0.226 | 0.246 | 0.265 |
| | | Freeway | Normal | 0.020 | 0.352 | 0.391 | 0.429 |
| | | others | Normal | 0.002 | 0.027 | 0.030 | 0.033 |

**Table S9. Uncertainty range of emission inventories.**

| | | Unit | Mean | Standard division | C.V | The 95% confidence interval | | |
|---|---|---|---|---|---|---|---|---|
| | | | | | | 2.5% percentile | 50 % percentile | 97.5% percentile |
| Tailpipe emissions | Passenger vehicles tailpipe emissions | Gg | 1279.12 | 252.51 | 0.20 | 902.39 | 1237.21 | 1891.96 |
| | Trucks tailpipe emissions | Gg | 720.89 | 45.20 | 0.06 | 636.52 | 718.39 | 816.43 |
| | Motorcycles tailpipe emissions | Gg | 562.54 | 349.17 | 0.62 | 158.61 | 476.40 | 1444.66 |
| Evaporative emissions | Diurnal emissions (excluding motorcycles) | Gg | 138.99 | 75.27 | 0.54 | 56.22 | 124.26 | 312.78 |
| | Hot Soak emissions (excluding motorcycles) | Gg | 15.75 | 3.71 | 0.24 | 9.70 | 15.33 | 24.26 |
| | Refueling emissions | Gg | 109.38 | 7.46 | 0.07 | 95.82 | 108.94 | 124.92 |
| | Running loss | Gg | 1146.18 | 768.92 | 0.67 | 229.90 | 963.11 | 3132.67 |
| | Motorcycles evaporation | Gg | 251.30 | 278.70 | 1.11 | 29.31 | 170.14 | 954.21 |
| Ratio of evaporative emissions versus tailpipe emissions of passenger cars | | | 1.14 | 0.67 | 0.5828 | 0.36 | 0.98 | 2.89 |
| Total emissions | | Gg | 4224.14 | 943.21 | 0.22 | 2897.14 | 4053.82 | 6540.95 |

Technical Corrections:

Section 3.2.2-3.2.4 are too short to be an individual section. I personally think that this part of discussion is not necessarily to be divided into three sections.

**Response:** Accepted. The original Sect. 3.2.1-3.2.4 were combined to Sect. 3.2. No sub-section was divided.

Supporting Information, Table S4: Some abbreviations of vehicle types (LDGTAs, LDDTAs) ought to be specified.

**Response:** Corrected. We have modified Sect. 2.1. Now the vehicle classification and abbreviations are consistent through the whole manuscript.

Some in-text citations are missing in the reference list, e.g., MOVES, 2010; ICCT, 2012.

**Response:** Corrected.